# Learning and Testing Causal Models with Interventions

**Jayadev Acharya**[*]
School of ECE
Cornell University
acharya@cornell.edu

**Arnab Bhattacharyya**[*]
National University of Singapore
& Indian Institute of Science
arnabb@iisc.ac.in

**Constantinos Daskalakis**[*]
EECS
MIT
costis@csail.mit.edu

**Saravanan Kandasamy**[*]
STCS
Tata Institute of Fundamental Research
saravan.tuty@gmail.com

## Abstract

We consider testing and learning problems on causal Bayesian networks as defined by Pearl [Pea09]. Given a causal Bayesian network $\mathcal{M}$ on a graph with $n$ discrete variables and bounded in-degree and bounded "confounded components", we show that $O(\log n)$ interventions on an unknown causal Bayesian network $\mathcal{X}$ on the same graph, and $O(n/\epsilon^2)$ samples per intervention, suffice to efficiently distinguish whether $\mathcal{X} = \mathcal{M}$ or whether there exists some intervention under which $\mathcal{X}$ and $\mathcal{M}$ are farther than $\epsilon$ in total variation distance. We also obtain sample/time/intervention efficient algorithms for: (i) testing the identity of two unknown causal Bayesian networks on the same graph; and (ii) learning a causal Bayesian network on a given graph. Although our algorithms are non-adaptive, we show that adaptivity does not help in general: $\Omega(\log n)$ interventions are necessary for testing the identity of two unknown causal Bayesian networks on the same graph, even adaptively. Our algorithms are enabled by a new subadditivity inequality for the squared Hellinger distance between two causal Bayesian networks.

## 1 Introduction

A central task in statistical inference is learning properties of a high-dimensional distribution over some variables of interest given observational data. However, probability distributions only capture the association between variables of interest and may not suffice to predict what the consequences would be of setting some of the variables to particular values. A standard example illustrating the point is this: From observational data, we may learn that atmospheric air pressure and the readout of a barometer are correlated. But can we predict whether the atmospheric pressure would stay the same or go up if the barometer readout was forcefully increased by moving its needle?

Such issues are at the heart of *causal inference*, where the goal is to learn a *causal model* over some variables of interest, which can predict the result of *external interventions* on the variables. For example, a causal model on two variables of interest $X$ and $Y$ need not only determine conditional probabilities of the form $\Pr[Y \mid X = x]$, but also *interventional probabilities* $\Pr[Y \mid do(X = x)]$

---

[*]The authors are arranged in alphabetical ordering. Jayadev Acharya was supported by a Cornell University startup grant. Arnab Bhattacharyya was partially supported by DST Ramanujan grant DSTO1358 and DRDO Frontiers Project DRDO0687. Constantinos Daskalakis was supported by NSF awards CCF-1617730 and IIS-1741137, a Simons Investigator Award, a Google Faculty Research Award, and an MIT-IBM Watson AI Lab research grant. Saravanan Kandasamy was partially supported by DRDO Frontiers Project DRDO0687.

where, following Pearl's notation [Pea09], $do(X = x)$ means that $X$ has been forced to take the value $x$ by an external action. In our previous example, $\Pr[\text{Pressure} \mid do(\text{Barometer} = b)] = \Pr[\text{Pressure}]$ but $\Pr[\text{Barometer} \mid do(\text{Pressure} = p)] \neq \Pr[\text{Barometer}]$, reflecting that the atmospheric pressure causes the barometer readout, not the other way around.

Causality has been the focus of extensive study, with a wide range of analytical frameworks proposed to capture causal relationships and perform causal inference. A prevalent class of causal models are *graphical causal models*, going back to Wright [Wri21] who introduced such models for path analysis, and Haavelmo [Haa43] who used them to define structural equation models. Today, graphical causal models are widely used to represent causal relationships in a variety of ways [SDLC93, GC99, Pea09, SGS00, Nea04, KF09].

In our work, we focus on the central model of *causal Bayesian networks* (CBNs) [Pea09, SGS00, Nea04]. Recall that a (standard) Bayesian network is a distribution over several random variables that is associated with a directed acyclic graph. The vertices of the graph are the random variables over which the distribution is defined, and the graph describes conditional independence properties of the distribution. In particular, every variable is independent of its non-descendants, conditioned on the values of its parents in the graph. A CBN is also associated with a directed acyclic graph (DAG) whose vertices are the random variables on which the distribution is defined. However, a CBN is not a single distribution over these variables but the collection of all possible interventional distributions, defined by setting any subset of the variables to any set of values. In particular, every vertex is both a variable $V$ and a *mechanism* to generate the value of $V$ given the values of the parent vertices, and the interventional distributions are defined in terms of these mechanisms.

We allow CBNs to contain both observable and unobservable (hidden) random variables. Importantly, we allow *unobservable confounding variables*. These are variables that are not observable, yet they are ancestors of at least two observable variables. These are especially tricky in statistical inference, as they may lead to spurious associations.

## 1.1 Our Contributions

Consider the following situations:

1. An engineer designs a large circuit using a circuit simulation program and then builds it in hardware. The simulator predicts relationships between the voltages and currents at different nodes of the circuit. Now, the engineer would like to verify whether the simulator's predictions hold for the real circuit by doing a limited number of experiments (e.g., holding some voltages at set levels, cutting some wires, etc.). If not, then she would want to learn a model for the system that has sufficiently good accuracy.
2. A biologist is studying the role of a set of genes in migraine. He would like to know whether the mechanisms relating the products of these genes are approximately the same for patients with and without migraine. He has access to tools (e.g., CRISPR-based gene editing technologies [DPL+16]) that generate data for gene activation and knockout experiments.

Motivated by such scenarios, we study the problems of hypothesis testing and learning CBNs when both observational and interventional data are available. The main highlight of our work is that we prove bounds on the number of samples, interventions, and time steps required by our algorithms.

To define our problems precisely, we need to specify what we consider to be a good approximation of a causal model. Given $\epsilon \in (0, 1)$, we say that two causal models $\mathcal{M}$ and $\mathcal{N}$ on a set of variables $\mathbf{V} \cup \mathbf{U}$ (observable and unobservable resp.) are $\epsilon$-*close* (denoted $\Delta(\mathcal{M}, \mathcal{N}) \leq \epsilon$) if for every subset $\mathbf{S}$ of $\mathbf{V}$ and assignment $\mathbf{s}$ to $\mathbf{S}$, performing the same intervention $do(\mathbf{S} = \mathbf{s})$ to both $\mathcal{M}$ and $\mathcal{N}$ leads to the two interventional distributions being $\epsilon$-close to each other in total variation distance. Otherwise, the two models are said to be $\epsilon$-*far* and $\Delta(\mathcal{M}, \mathcal{N}) > \epsilon$.

Thus, two models $\mathcal{M}$ and $\mathcal{N}$ are close according to the above definition if there is *no* intervention which can make the resulting distributions differ significantly. This definition is motivated by the philosophy articulated by Pearl (pp. 414, [Pea09]) that "causation is a summary of behavior under intervention". Intuitively, if there is some intervention that makes $\mathcal{M}$ and $\mathcal{N}$ behave differently, then $\mathcal{M}$ and $\mathcal{N}$ do not describe the same causal process. Without having any prior information about the

set of relevant interventions, we adopt a worst-case view and simply require that causal models $\mathcal{M}$ and $\mathcal{N}$ behave similarly for every intervention to be declared close to each other.[2]

The *goodness-of-fit testing* problem can now be described as follows. Suppose that a collection $\mathbf{V} \cup \mathbf{U}$ (observable and unobservable resp.) of $n$ random variables are causally related to each other. Let $\mathcal{M}$ be a hypothesized causal model for $\mathbf{V} \cup \mathbf{U}$ that we are given explicitly. Suppose that the true model to describe the causal relationships is an unknown $\mathcal{X}$. Then, the goodness-of-fit testing problem is to distinguish between: (i) $\mathcal{X} = \mathcal{M}$, versus (ii) $\Delta(\mathcal{X}, \mathcal{M}) > \epsilon$, by sampling from and experimenting on $\mathbf{V}$, i.e. forcing some variables in $\mathbf{V}$ to certain values and sampling from the thus intervened upon distribution.

We study goodness-of-fit testing assuming $\mathcal{X}$ and $\mathcal{M}$ are causal Bayesian networks over a known DAG $G$. Given a DAG $G$, CBN $\mathcal{M}$ and $\epsilon > 0$, we denote the corresponding goodness-of-fit testing problem $\mathsf{CGFT}(G, \mathcal{M}, \epsilon)$. For example, the engineer above, who wants to determine whether the circuit behaves as the simulation software predicts, is interested in the problem $\mathsf{CGFT}(G, \mathcal{M}, \epsilon)$ where $\mathcal{M}$ is the simulator's prediction, $G$ is determined by the circuit layout, and $\epsilon$ is a user-specified accuracy parameter. Here is our theorem for goodness-of-fit testing.

**Theorem 1** (Goodness-of-fit Testing – Informal). *Let $G$ be a DAG on $n$ vertices with bounded in-degree and bounded "confounded components." Let $\mathcal{M}$ be a given CBN over $G$. Then, there exists an algorithm solving $\mathsf{CGFT}(G, \mathcal{M}, \epsilon)$ that makes $O(\log n)$ interventions, takes $O(n/\epsilon^2)$ samples per intervention and runs in time $\tilde{O}(n^2/\epsilon^2)$. Namely, the algorithm gets access to a CBN $\mathcal{X}$ over $G$, accepts with probability $\geq 2/3$ if $\mathcal{X} = \mathcal{M}$ and rejects with probability $\geq 2/3$ if $\Delta(\mathcal{X}, \mathcal{M}) > \epsilon$.*

By "confounded component" in the above statement, we mean a *c-component* in $G$, as defined in Definition 7. Roughly, a c-component is a maximal set of observable vertices that are pairwise connected by paths of the form $V_{i_1} \leftarrow U_{j_1} \rightarrow V_{i_2} \leftarrow U_{j_2} \rightarrow V_{i_3} \leftarrow \cdots \rightarrow V_{i_k}$ where $V_i$'s and $U_j$'s correspond to observable and unobservable variables respectively. The decomposition of CBNs into c-components has been important in earlier work [TP02] and continues to be an important structural property here.

We can use our techniques to extend Theorem 1 in several ways:

(1) In the *two-sample testing* problem for causal models, the tester gets access to two unknown causal models $\mathcal{X}$ and $\mathcal{Y}$ on the same set of variables $\mathbf{V} \cup \mathbf{U}$ (observable and unobservable resp.). For a given $\epsilon > 0$, the goal is to distinguish between (i) $\mathcal{X} = \mathcal{Y}$ and (ii) $\Delta(\mathcal{X}, \mathcal{Y}) > \epsilon$ by sampling from and intervening on $\mathbf{V}$ in both $\mathcal{X}$ and $\mathcal{Y}$.
     We solve the two-sample testing problem when the inputs are two CBNs over the same DAG $G$ in $n$ variables; for a given $\epsilon > 0$ and DAG $G$, call the problem $\mathsf{C2ST}(G, \epsilon)$. Specifically, we show an algorithm to solve $\mathsf{C2ST}(G, \epsilon)$ that makes $O(\log n)$ interventions on the input models $\mathcal{X}$ and $\mathcal{Y}$, uses $O(n/\epsilon^2)$ samples per intervention and runs in time $\tilde{O}(n^2/\epsilon^2)$, when $G$ has bounded in-degree and c-component size.[3]
(2) For the $\mathsf{C2ST}(G, \epsilon)$ problem, the requirement that $G$ be fully known is rather strict. Instead, suppose the common graph $G$ is unknown and only bounds on its in-degree and maximum c-component size are given. For example, the biologist above who wants to test whether certain causal mechanisms are identical for patients with and without migraine can reasonably assume that the underlying causal graph is the same (even though he doesn't know what it is exactly) and that only the strengths of the relationships may differ between subjects with and without migraine. For this problem, we obtain an efficient algorithm with nearly the same number of samples and interventions as above.
(3) The problem of *learning* a causal model can be posed as follows: the learning algorithm gets access to an unknown causal model $\mathcal{X}$ over a set of variables $\mathbf{V} \cup \mathbf{U}$ (observable and unobservable resp.), and its objective is to output a causal model $\mathcal{N}$ such that $\Delta(\mathcal{X}, \mathcal{N}) \leq \epsilon$. We consider the problem $\mathsf{CL}(G, \epsilon)$ of learning a CBN over a known DAG $G$ on the observable and unobservable variables. For example, this is the problem facing the engineer above

who wants to learn a good model for his circuit by conducting some experiments; the DAG $G$ in this case is known from the circuit layout. Given a DAG $G$ with bounded in-degree and c-component size and a parameter $\epsilon > 0$, we design an algorithm that on getting access to a CBN $\mathcal{X}$ defined over $G$, makes $O(\log n)$ interventions, uses $\tilde{O}(n^2/\epsilon^4)$ samples per intervention, runs in time $\tilde{O}(n^3/\epsilon^4)$, and returns an oracle $\mathcal{N}$ that can efficiently compute $P_{\mathcal{X}}[\mathbf{V} \setminus \mathbf{T} \mid do(\mathbf{T} = \mathbf{t})]$ for any $\mathbf{T} \subseteq \mathbf{V}$ and $\mathbf{t} \in \Sigma^{|\mathbf{T}|}$ with error at most $\epsilon$ in TV distance.

The sample complexity of our testing algorithms matches the state-of-the-art for testing identity of (standard) Bayes nets [DP17, CDKS17]. Designing a goodness-of-fit tester using $o(n)$ samples is a very interesting challenge and seems to require fundamentally new techniques.

We also show that the number of interventions for $\mathsf{C2ST}(G, \epsilon)$ and $\mathsf{CL}(G, \epsilon)$ is nearly optimal, even in its dependence on the in-degree and c-component size, and even when the algorithms are allowed to be adaptive. By 'adaptive' we mean the algorithms are allowed to choose the future interventions based on the samples observed from the past interventions. Specifically,

**Theorem 2.** *There exists a causal graph $G$ on $n$ vertices, with maximum in-degree at most $d$ and largest c-component size at most $\ell$, such that $\Omega(|\Sigma|^{\ell d - 2} \log n)$ interventions are necessary for any algorithm (even adaptive) that solves $\mathsf{C2ST}(G, \epsilon)$ or $\mathsf{CL}(G, \epsilon)$, where $\Sigma$ is the alphabet set from which the variables take values.*

We make no assumptions about the distributions or the functional relationships, and we show that in the worst case, the $K^{\ell d}$ bound, that appears in the number of interventions, is unavoidable. However, with further assumptions, one can hope to reduce this number. For example, if the graph has no hidden variables and each assignment to the parent sets occurs with large enough probability, access to interventions is not necessary. Or if the mechanism relating each variable to its parents can be modeled as a linear function, then a covering set of interventions is not needed.

## 1.2 Related Work

(A longer discussion of previous work on causality as well as on testing/learning distributions is in Appendix A.) There is a huge and old literature on causality, for both and learning testing causal relationships that is impossible to detail here. To the best of our knowledge, though, most previous work is on testing/learning only the causal graph, whereas our objective is to test/learn the entire causal model (i.e., the set of all interventional distributions). In fact, many of our results assume that the causal graph is already known; as discussed in Section 1.4, we hope that in future work, this requirement can be relaxed.

Motivated by the problem of testing causal graphs, Tian and Pearl [TP02] derive functional constraints among the distributions of observed variables (not just conditional independence relations) in a causal Bayesian network over the graph. Kang and Tian [KT06] derive such functional constraints on interventional distributions. Although these results yield non-trivial constraints, it is not clear how to use them for testing goodness-of-fit with statistical guarantees.

The problem of learning causal graphs has been extensively studied. [PV95, VP92, SGS00, ARSZ05, Zha08] give algorithms to recover the class of causal graphs consistent with given conditional independence relations in observational data. Subsequent work considered the setting when both observational and interventional data are available. This setting has been a recent focus of study [HB12a, WSYU17, YKU18], motivated by advances in genomics that allow high-resolution observational and interventional data for gene expression using flow cytometry and CRISPR technologies [SPP+05, MBS+15, DPL+16]. [EGS05, Ebe07, HB12b] derived the minimum number of interventional distributions that suffice to fully identify the underlying causal graphs when there are no confounding variables. Recently, Kocaoglu et al. [KSB17] showed an efficient randomized algorithm to learn a causal graph with confounding variables while minimizing the number of interventions from which conditional independence relations are obtained.

From the perspective of query learning, learning circuits with value injection queries was introduced by Angluin et al. [AACW09]. The value injection query model is a deterministic circuit defined over an underlying directed acyclic graph whose output is determined by the value of the output node. [AACW09] considers the problem of learning the outputs of all value injection queries (i.e., interventions) where the learner has oracle access to value injection queries with the objective of

minimizing the number of queries, when the size of alphabet set is constant. This was later generalized to large alphabet and analog circuits in [AACR08, Rey09].

All the works mentioned above assume access to an oracle that gives conditional independence relations between variables in the observed and interventional distributions. This is clearly a problematic assumption because it implicitly requires unbounded training data. For example, Scheines and Spirtes [SS08] have pointed out that measurement error, quantization and aggregation can easily alter conditional independence relations. The problem of developing finite sample bounds for testing and learning causal models has been repeatedly posed in the literature. The excellent survey by Guyon, Janzing and Schölkopf [GJS10] on causality from a machine learning perspective underlines the issue as one of the "ten open problems" in the area. To the best of our knowledge, our work is the first to show finite sample complexity and running time bounds for inference problems on CBNs.

An application of our learning algorithm is to the problem of *transportability*, studied in [BP13, SP08, LH13, PB11, BP12], which refers to the notion of transferring causal knowledge from a set of source domains to a target domain to identify causal effects in the target domain, when there are certain commonalities between the source and target domains. Most work in this area assume the existence of an algorithm that learns the set of *all* interventions, that is the complete specification of the source domain model. Our learning algorithm can be used for this purpose; it is efficient in terms of time, interventions, and sample complexity, and it learns each intervention distribution to error at most $\epsilon$.

## 1.3  Overview of our Techniques

In this section, we give an overview of the proof of Theorem 1 and the lower bound construction. We start by making a well-known observation [TP02, VP90] that CBNs can be assumed to be over a particular class of DAGs known as *semi-Markovian causal graphs*. A semi-Markovian causal graph is a DAG where every vertex corresponding to an unobservable variable is a root and has exactly two children, both observable. More details of the correspondence are given in Appendix I.

In a semi-Markovian causal graph, two observable vertices $V_1$ and $V_2$ are said to be connected by a bi-directed edge if there is a common unobservable parent of $V_1$ and $V_2$. Each connected component of the graph restricted to bi-directed edges is called a *c-component*. The decomposition into c-components forms a partition of the observable vertices, which gives very useful structural information about the causal model. In particular, a fact that is key to our whole analysis is that if $\mathcal{N}$ is a semi-Markovian Bayesian network on observable and unobservable variables $\mathbf{V} \cup \mathbf{U}$ with c-components $\mathbf{C}_1, \ldots, \mathbf{C}_p$, then for any $\mathbf{v} \in \Sigma^{|\mathbf{V}|}$:

$$P_\mathcal{N}[\mathbf{v}] = \prod_{i=1}^{p} P_\mathcal{N}[\mathbf{c}_i \mid do(\mathbf{V} \setminus \mathbf{C}_i = \mathbf{v} \setminus \mathbf{c}_i)] \tag{1}$$

where $\Sigma$ is the alphabet set, $\mathbf{c}_i$ is the restriction of $\mathbf{v}$ to $\mathbf{C}_i$ and $\mathbf{v} \setminus \mathbf{c}_i$ is the restriction of $\mathbf{v}$ to $\mathbf{V} \setminus \mathbf{C}_i$ [TP02]. Moreover, one can write a similar formula (Lemma 9) for an interventional distribution on $\mathcal{N}$ instead of the observable distribution $P_\mathcal{N}[\mathbf{v}]$.

The most direct approach to test whether two causal Bayes networks $\mathcal{X}$ and $\mathcal{Y}$ are identical is to test whether each interventional distribution is identical in the two models. This strategy would require $(|\Sigma| + 1)^n$ many interventions, each on a variable set of size $O(n)$, where $n$ is the total number of observable vertices. To reduce the number of interventions as well as the sample complexity, a natural approach, given (1) and its extension to interventional distributions, is to test for identity between each pair of "local" distributions

$$P_\mathcal{X}[\mathbf{S} \mid do(\mathbf{v} \setminus \mathbf{s})] \qquad \text{and} \qquad P_\mathcal{Y}[\mathbf{S} \mid do(\mathbf{v} \setminus \mathbf{s})]$$

for every subset $\mathbf{S}$ of a c-component $\mathbf{C}$ and assignment $\mathbf{v} \setminus \mathbf{s}$ to $\mathbf{V} \setminus \mathbf{S}$. We assume that each c-component is bounded, so each local distribution has bounded support. Moreover, using the conditional independence properties of Bayesian networks, note that in each local distribution, we only need to intervene on observable parents of $\mathbf{S}$ that are outside $\mathbf{S}$, not on all of $\mathbf{V} \setminus \mathbf{S}$.

Through a probabilistic argument, we efficiently find a *small set* $\mathbf{I}$ of *covering interventions*, which are defined as a set of interventions with the following property: For every subset $\mathbf{S}$ of a c-component and for every assignment $\mathbf{pa}(\mathbf{S})$ to the observable parents of $\mathbf{S}$, there is an intervention $I \in \mathbf{I}$ that does not intervene on $\mathbf{S}$ and sets the parents of $\mathbf{S}$ to exactly $\mathbf{pa}(\mathbf{S})$. Our test performs all the interventions in $\mathbf{I}$ on both $\mathcal{X}$ and $\mathcal{Y}$ and hence can observe each of the local distributions $P_\mathcal{X}[\mathbf{S} \mid do(\mathbf{pa}(\mathbf{S}))]$ and

$P_{\mathcal{Y}}[\mathbf{S} \mid do(\mathbf{pa}(\mathbf{S}))]$. What remains is to bound $\Delta(\mathcal{X}, \mathcal{Y})$ in terms of the distances between each pair of local distributions.

To that end, we develop a subadditivity theorem about CBNs, and this is the main technical contribution of our upper bound results. We show that if each pair of local distributions is within distance $\gamma$ in *squared Hellinger* distance, then for any intervention $I$, applying $I$ to $\mathcal{X}$ and $\mathcal{Y}$ results in distributions that are within $O(n\gamma)$ distance in squared Hellinger distance, assuming bounded in-degree and c-component size of the underlying graph. A bound on the total variation distance between the interventional distributions and hence $\Delta(\mathcal{X}, \mathcal{Y})$ follows. The subadditivity theorem is inspired from [DP17], where they showed that for Bayes networks, "closeness of local marginals implies closeness of the joint distribution". Our result is in a very different set-up, where we prove "closeness of local interventions implies closeness of any joint interventional distribution", and requires a new proof technique. We relax the squared Hellinger distance between the interventional distributions as the objective of a minimization program in which the constraints are that each pair of local distributions is $\gamma$-close in squared Hellinger distance. By a sequence of transformations of the program, we lower bound its objective in terms of $\gamma$, thus proving our result. In the absence of unobservable variables, the analysis becomes much simpler and is sketched in Appendix B.

Regarding the lower bound, we prove that the number of interventions required by our algorithms are indeed necessary for any algorithm that solves $\mathsf{C2ST}(G, \epsilon)$ or $\mathsf{CL}(G, \epsilon)$, even if the algorithms are provided with infinite samples/time. For any algorithm that fails to perform some local intervention $I$, we provide a construction of two models which do not agree on $I$ and agree on all other interventions. Our construction is designed in such a way that it allows adaptive algorithms. The idea is to show an adversary that, for each intervention, reveals a distribution to the algorithm. Towards the end, when the algorithm fails to perform some local intervention $I$, we can show a construction of two models such that: i) both the models do not agree on $I$, and the total variation distance between the interventional distributions is equal to one; ii) and for all other interventions, the interventional distributions revealed by the adversary match with the corresponding distributions on both the models. This, together with a probabilitic argument, shows the existence of a causal graph that requires sufficiently large number of interventions to solve $\mathsf{C2ST}(G, \epsilon)$ and $\mathsf{CL}(G, \epsilon)$.

## 1.4 Future Directions

We hope that this work paves the way for future research on designing efficient algorithms with bounded sample complexity for learning and testing causal models. For the sake of concreteness, we list a few open problems.

- Interventional experiments are often expensive or infeasible, so one would like to deduce causal models from observations alone. In general, this is impossible. However, in *identifiable* CBNs (see [Tia02]), one can identify causal effects from observational data alone. **Is there an efficient algorithm to learn an identifiable interventional distribution from samples?**[4]

- A deficiency of our learning algorithm is that we assume the underlying causal graph is fully known. **Can our learning algorithm be extended to the setting where the hypothesis only consists of some limited information about the causal graph (e.g., in-degree, c-component size) instead of the whole graph?** This seems to be a hard problem. In fact, it is open how to efficiently learn the distribution given by a standard Bayesian network based on samples from it if we don't know the underlying graph [DP17, CDKS17].

- Our goodness-of-fit algorithm might reject even when the input $\mathcal{X}$ is very close to the hypothesis $\mathcal{M}$. **Is there a *tolerant* goodness-of-fit tester that accepts when $\Delta(\mathcal{X}, \mathcal{M}) \leq \epsilon_1$ and rejects when $\Delta(\mathcal{X}, \mathcal{M}) > \epsilon_2$ for $0 < \epsilon_1 < \epsilon_2 < 1$?** Our current analysis does not extend to a tolerant tester. The same question holds for testing goodness-of-fit for standard Bayesian networks.

- In many applications, causal models are described in terms of *structural equation models*, in which each variable is a deterministic function of its parents as well as some stochastic error terms. **Design sample and time efficient algorithms for testing and learning structural equation**

**models.** Other questions such as evaluating *counterfactual* queries or doing *policy analysis* (see Chapter 7 of [Pea09]) also present interesting algorithmic problems.

## 2 Testing and Learning Algorithms for SMBNs

We use SMCG and SMBN to denote semi-Markovian causal graph and semi-Markovian Bayesian network respectively on $\mathbf{V} \cup \mathbf{U}$, observable and unobservable variables respectively. Let $\mathcal{G}_{d,\ell}$ denotes the class of SMCGs with maximum in-degree at most $d$ and largest c-component size at most $\ell$. For any subset $\mathbf{S}$ of observable variables, we use $\mathbf{Pa}(\mathbf{S})$ to denote the observable parents of $\mathbf{S}$ (excluding $\mathbf{S}$), and $\mathbf{pa}(\mathbf{S})$ to denote an assignment to $\mathbf{Pa}(\mathbf{S})$. More formal definitions can be found in Appendix C.

First we recall a fast and sample-efficient test for squared Hellinger distance from [DKW18].

**Lemma 1.** *[Hellinger Test, [DKW18]] Given $O(\min(D^{2/3}/\epsilon^{8/3}, D^{3/4}/\epsilon^2))$ samples from each unknown distributions $P$ and $Q$, we can distinguish between $P = Q$ vs $H^2(P,Q) \geq \epsilon^2$ with probability at least $2/3$. This probability can be boosted to $1 - \delta$ at a cost of an additional $O(\log(1/\delta))$ factor in the sample complexity. The running time of the algorithm is quasi-linear in the sample size [5].*

We also need the notion of *covering intervention sets*:

**Definition 1.** *A set of interventions $\mathbf{I}$ is a* covering intervention set *if for every subset $\mathbf{S}$ of every c-component, and every assignment $\mathbf{pa}(\mathbf{S}) \in \Sigma^{|\mathbf{Pa}(\mathbf{S})|}$ there exists an $I \in \mathbf{I}$ such that, (i) No node in $\mathbf{S}$ is intervened in $I$; (ii) Every node in $\mathbf{Pa}(S)$ is intervened; and (iii) $I$ restricted to $\mathbf{Pa}(S)$ has the assignment $\mathbf{pa}(S)$.*

Our algorithms comprise of two key arguments.

- A procedure to compute a covering intervention set $\mathbf{I}$ of *small size*, given as Lemma 2 below.

- A sub-additivity result, shown in Theorem 3, for CBNs that allows us to localize the distances: where we show that two CBNs are far implies there exist a marginal distribution of some intervention in $\mathbf{I}$ such that the marginals are far.

**Lemma 2.** (Counting Lemma) *Let $G \in \mathcal{G}_{d,\ell}$ be a SMCG with $n$ vertices and $\Sigma$ be an alphabet set of size $K$. Then, there exists a covering intervention set $\mathbf{I}$ of size $O(K^{\ell d}(3d)^\ell(\log n + \ell d \log K))$. If the total degree of $G$ is bounded by $d$, then there exists such an $\mathbf{I}$ of size $O(K^{\ell d}(3d)^\ell \ell d^2 \log K)$. In both cases, there is an $\tilde{O}(n)$ time algorithm to output $\mathbf{I}$.*

**Theorem 3.** (Subadditivity Theorem) *Let $\mathcal{M}$ and $\mathcal{N}$ be two SMBNs defined on a known and common SMCG $G \in \mathcal{G}_{d,\ell}$. For a given intervention $do(\mathbf{t})$, let $\mathbf{V} \setminus \mathbf{T}$ partition into $\mathcal{C} = \{\mathbf{C}_1, \mathbf{C}_2, \ldots, \mathbf{C}_p\}$, the c-components with respect to the induced graph $G[\mathbf{V} \setminus \mathbf{T}]$. Suppose*

$$H^2(P_\mathcal{M}[\mathbf{C}_j \mid do(\mathbf{pa}(\mathbf{C}_j))], P_\mathcal{N}[\mathbf{C}_j \mid do(\mathbf{pa}(\mathbf{C}_j))]) \leq \gamma \qquad \forall j \in [p], \forall \mathbf{pa}(\mathbf{C}_j) \in \Sigma^{|\mathbf{Pa}(\mathbf{C}_j)|}. \tag{2}$$

*Then*

$$H^2\left(P_\mathcal{M}[\mathbf{V} \setminus \mathbf{T} \mid do(\mathbf{t})], P_\mathcal{N}[\mathbf{V} \setminus \mathbf{T} \mid do(\mathbf{t})]\right) \leq \epsilon \qquad \forall \mathbf{t} \in \Sigma^{|\mathbf{T}|} \tag{3}$$

*where $\epsilon = \gamma |\Sigma|^{\ell(d+1)} n$.*

The proof of Lemma 2 is shown in Appendix E.1. The subadditivity theorem is proved in Appendix E.2.

Our main testing algorithm for $\mathsf{C2ST}(G, \epsilon)$ is shown below in Theorem 4, which gives Theorem 1 as a corollary, since two sample tests are harder than one sample tests. We also provide 1) an algorithm for $\mathsf{C2ST}(G, \epsilon)$ when $G \in \mathcal{G}_{d,\ell}$ is unknown, and 2) an algorithm for $\mathsf{CL}(G, \epsilon)$. Both these algorithms are similar to the below algorithm and can be found in Appendix D.

**Theorem 4** (Algorithm for $\mathsf{C2ST}(G, \epsilon)$)**.** *Let $G$ be a SMCG $\in \mathcal{G}_{d,\ell}$ with $n$ vertices. Let the variables take values over a set $\Sigma$ of size $K$. Then, there is an algorithm to solve $\mathsf{C2ST}(G, \epsilon)$, that makes $O(K^{\ell d}(3d)^{\ell} \log n)$ interventions to each of the unknown SMBNs $\mathcal{X}$ and $\mathcal{Y}$, taking $O(K^{\ell(d+7/4)} n \epsilon^{-2})$ samples per intervention, in time $\tilde{O}(2^{\ell} K^{\ell(2d+7/4)} n^2 \epsilon^{-2})$.*

*When the maximum degree (in-degree plus out-degree) of $G$ is bounded by $d$, then our algorithm uses $O(K^{\ell d}(3d)^{\ell} \ell d^2 \log K)$ interventions with the same sample complexity and running time as above.*

*Proof of Theorem 4.* Our algorithm is described in Algorithm 1. The algorithm starts with a covering intervention set $\mathbf{I}$. Lemma 2 gives an $\mathbf{I}$ with $O(K^{\ell d}(3d)^{\ell}(\log n + \ell d \log K))$ interventions., and when the maximum degree is bounded by $d$, then the same lemma gives an $\mathbf{I}$ of size $O(K^{\ell d}(3d)^{\ell} \ell d^2 \log K)$.

---

**Algorithm 1**: Algorithm for $\mathsf{C2ST}(G, \epsilon)$

$\mathbf{I}$: Covering intervention set

1. Under each intervention $I \in \mathbf{I}$:

   (a) Obtain $O(K^{\ell(d+7/4)} n \epsilon^{-2})$ samples from the interventional distribution of $I$ in both models $\mathcal{X}$ and $\mathcal{Y}$.

   (b) For any subset $\mathbf{S}$ of a c-component of $G$, if $I$ does not set $\mathbf{S}$ but sets $\mathbf{Pa}(\mathbf{S})$ to $\mathbf{pa}(\mathbf{S})$, then using Lemma 1 and the obtained samples, test (with error probability at most $1/(3K^{\ell d} 2^{\ell} n)$):
   $$P_{\mathcal{X}}[\mathbf{S}|do(\mathbf{pa}(\mathbf{S}))] = P_{\mathcal{Y}}[\mathbf{S}|do(\mathbf{pa}(\mathbf{S}))] \text{ vs } H^2 \left( \begin{array}{c} P_{\mathcal{X}}[\mathbf{S}|do(\mathbf{pa}(\mathbf{S}))], \\ P_{\mathcal{Y}}[\mathbf{S}|do(\mathbf{pa}(\mathbf{S}))] \end{array} \right) \geq \frac{\epsilon^2}{2K^{\ell(d+1)} n}.$$
   Output "$\Delta(\mathcal{X}, \mathcal{Y}) > \epsilon$" if the latter.

2. Output "$\mathcal{X} = \mathcal{Y}$".

---

We will now analyze the performance of our algorithm.

**Number of interventions, time, and sample requirements.** The number of interventions is the size of $\mathbf{I}$, bounded above. The number of samples per intervention is given in the algorithm. The algorithm performs $n2^{\ell} K^{\ell d}$ sub-tests. And for each such sub-test, the algorithm's running time is quasi-linear in the sample complexity (Lemma 1), therefore taking a total time of $\tilde{O}(2^{\ell} K^{\ell(2d+7/4)} n^2 \epsilon^{-2})$.

**Correctness.** In Theorem 3, we show that when $\Delta(\mathcal{X}, \mathcal{Y}) > \epsilon$, there exists a subset $\mathbf{S}$ of some c-component, and an $I \in \mathbf{I}$ that does not intervene any node in $\mathbf{S}$ but intervenes $\mathbf{Pa}(\mathbf{S})$ with some assignment $\mathbf{pa}(\mathbf{s})$ such that

$$H^2(P_{\mathcal{X}}[\mathbf{S} \mid do(\mathbf{pa}(\mathbf{S}))], P_{\mathcal{Y}}[\mathbf{S} \mid do(\mathbf{pa}(\mathbf{S}))]) > \epsilon^2/(2K^{\ell(d+1)} n).$$

This structural result is the key to our algorithm. This, together with the fact that the TV distance between two distributions is at most $\sqrt{2}$ times their Hellinger distance, proves that $P_{\mathcal{X}}$ and $P_{\mathcal{Y}}$ are far in terms of the total variation distance. To bound the error probability, note that the number of total sub-tests we run is bounded by $K^{\ell d} n 2^{\ell}$, and the error probability for each subset is at most $1/(3K^{\ell d} 2^{\ell} n)$, by the union bound, we will have an error of at most $1/3$ over the entire algorithm. $\quad\square$

## 3 Lower Bound on Interventional Complexity

Recall that in Section 2 we provided non-adaptive algorithms for $\mathsf{C2ST}(G, \epsilon)$, and $\mathsf{CL}(G, \epsilon)$. In this section we provide lower bounds on the number of interventions that any algorithm must make to solve these problems. Our lower bounds nearly match the upper bounds in Theorem 4, and Theorem 8, even when the algorithm is allowed to be adaptive (namely future interventions are decided based upon the samples observed from the past interventions). In other words, these lower bounds show that in general, adaptivity cannot reduce the interventional complexity.

**Theorem 5.** *There exists a SMCG $G \in \mathcal{G}_{d,\ell}$ with $n$ nodes such that $\Omega(K^{\ell d-2} \log n)$ interventions are necessary for any algorithm (even adaptive) that solves $\mathsf{C2ST}(G, \epsilon)$ or $\mathsf{CL}(G, \epsilon)$.*

This theorem is proved via the following ingredients.

**Necessary Condition.** We obtain a necessary condition on the set of interventions $\mathbf{I}$ of any algorithm that solves $\mathsf{C2ST}(G, \epsilon)$ or $\mathsf{CL}(G, \epsilon)$.

We will consider SMCGs $G$ with a specific structure, and prove the necessary condition for these graphs: The vertices of $G$ are the union of two disjoint sets $\mathbf{A}$, and $\mathbf{B}$, such that $G$ contains directed edges from $\mathbf{A}$ to $\mathbf{B}$, and bidirected edges within $\mathbf{B}$. Further, all edges in $G$ are one of these two types. The next lemma is for graphs with this structure.

**Lemma 3.** *Suppose an adaptive algorithm uses a sequence of interventions $\mathbf{I}$ to solve $\mathsf{C2ST}(G, \epsilon)$ or $\mathsf{CL}(G, \epsilon)$. Let $\mathbf{C} \subseteq \mathbf{B}$ be a c-component of $G$. Then, for any assignment $\mathbf{pa}(\mathbf{C}) \in \Sigma^{|\,\mathbf{Pa}(\mathbf{C})|}$, there is an intervention $I \in \mathbf{I}$ such that the following conditions hold:*

    **C1.** *$I$ intervenes $\mathbf{Pa}(\mathbf{C})$ with the corresponding assignment of $\mathbf{pa}(\mathbf{C})$,[6]*

    **C2.** *$I$ does not intervene on any node in $\mathbf{C}$.*

**Existence.** We then show that there is a graph with the structure mentioned above for which $\mathbf{I}$ must be $\Omega(K^{\ell d-2} \log n)$ in order for the condition to be satisfied. More precisely,

**Lemma 4.** *There exists a $G$, and a constant $c$ such that for any set of interventions $\mathbf{I}$ with $|\mathbf{I}| < c \cdot K^{\ell d-2} \log n$, there is a $\mathbf{C} \subseteq \mathbf{B}$, which is a c-component of $G$, and an assignment $\mathbf{pa}(\mathbf{C})$ such that no intervention in $\mathbf{I}$*

    • *assigns $\mathbf{pa}(\mathbf{C})$ to $\mathbf{Pa}(\mathbf{C})$, and*

    • *observes all variables in $\mathbf{C}$.*

Combining these two lemmas, we obtain the lower bound for the adaptive versions of $\mathsf{C2ST}(G, \epsilon)$ and $\mathsf{CL}(G, \epsilon)$. The proofs of Lemmas 3 and 4 are described in Appendix F.

**Acknowledgments**

We would like to thank Vasant Honavar who told us about the problems considered here and for several helpful discussions that were essential for us to complete this work. We acknowledge the support of Google India and NeurIPS in the form of an International Travel Grant, which enabled Saravanan Kandasamy to attend the conference.

## Footnotes

[2]To quote Pearl again, "It is the nature of any causal explanation that its utility be proven not over standard situations but rather over novel settings that require innovative manipulations of the standards." (pp. 219, [Pea09]).

[3]Of course, it is allowed for the two networks to be different subgraphs of $G$. So, $\mathcal{X}$ could be defined by the graph $G_1$ and $\mathcal{Y}$ by $G_2$. Our result holds when $G_1 \cup G_2$ is a DAG with bounded in-degree and c-component size.

[4]Schulman and Srivastava [SS16] have shown that under adversarial noise, there exist causal Bayesian networks on $n$ nodes where estimating an identifiable intervention to precision $d$ requires precision $d + \exp(n^{0.49})$ in the estimates of the probabilities of observed events. However, this instability is likely due to the adversarial noise and does not preclude an efficient sampling-based algorithm, especially if we assume a balancedness condition as in [CDKS17].

[5] The sample complexity here is an improvement of the previously known result of [DK16].

[6]In our construction, $\mathbf{Pa}(C)$ always take $\mathbf{0}$ in the natural distribution. Henceforth, the interventions where some vertices in $\mathbf{Pa}(C)$ are not intervened are not considered here, as they are equivalent to the case when those vertices are intervened with $\mathbf{0}$.

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
