[Supplementary Material]

# A   Related Work

## A.1   Causality

As mentioned before, there is a huge and old literature on causality, for both testing causal relationships and inferring causal graphs that is impossible to detail here. Below, we point out some representative directions of research that are relevant to our work. This discussion is far from exhaustive, and the reader is encouraged to pursue the references cited in the mentioned works.

Most work on statistical tests for causal models has been in the parametric setting. *Structural equation models* have traditionally been tested for goodness-of-fit by comparing observed and predicted covariance matrices [BL92]. Another class of tests that has been proposed assumes that the causal factors and the noise factors are conditionally independent. In the *additive noise model* [HJM$^+$09, PJS11, ZPJS12, SSS$^+$17], each variable is the sum of a (non-linear) function of its parent variables and independent noise, often assumed to be Gaussian. This point of view has been refined into an information-geometric criterion in [JMZ$^+$12]. In the non-parametric setting, which is the concern of this paper, Tian and Pearl [TP02] show how to derive functional constraints from causal Bayesian networks that give equality and inequality constraints among the (distributions of) observed variables, not just conditional independence relations. Kang and Tian [KT06] derive such functional constraints on interventional distributions. Although these results yield non-trivial constraints, it is not clear how to use them for testing goodness-of-fit with statistical guarantees.

Pearl and Verma [PV95, VP92] investigated whether a given list of conditional independence relations in observational data can be explained by a Bayesian net with unobserved variables. In fact, there may be a large number of causal Bayesian networks that are consistent with a given set of conditional independence relations. [SGS00, ARSZ05], and Zhang [Zha08] (building on the FCI algorithm [SMR99]) has given a complete and sound algorithm for recovering a representative of the equivalence class consistent with a set of conditional independence relations.

The problem of learning causal graphs has been extensively studied. When there are no confounding variables, Hauser and Bühlmann [HB12b], following up on work by Eberhardt and others [EGS05, Ebe07], find the information-theoretically minimum number of interventions that are sufficient to identify[7] the underlying causal graph and provide a polynomial time algorithm to find such a set of interventions. Subsequent work considered the setting when both observational and interventional data are available. Learning from the interventional setting has been a recent focus of study [HB12a, WSYU17, YKU18], motivated by advances in genomics that allow high-resolution observational and interventional data for gene expression using flow cytometry and CRISPR technologies [SPP$^+$05, MBS$^+$15, DPL$^+$16]. A recent paper [KDV17] extends the work of [HB12b] to minimize the total cost of interventions where each vertex is assigned a cost. Another work by Shanmugam et al. [SKDV15] investigates the problem of learning causal graphs without confounding variables using interventions on sets of small size. In the presence of confounding variables, there are several works which aim to learn the causal graph from interventional data (e.g., [MMLM06, HEH13]). In particular, a recent work of Kocaoglu et al. [KSB17] gives an efficient randomized algorithm to learn a causal network with confounding variables while minimizing the number of interventions from which conditional independence relations are obtained.

From the perspective of query learning, learning circuits with value injection queries was introduced by Angluin et al. [AACW09]. The value injection query model is a deterministic circuit defined over an underlying directed acyclic graph whose output is determined by the value of the output node. [AACW09] considers the problem of learning the outputs of all value injection queries (i.e., interventions) where the learner has oracle access to value injection queries with the objective of minimizing the number of queries, when the size of alphabet set is constant. This was later generalized to large alphabet and analog circuits in [AACR08, Rey09].

All the works mentioned above assume access to an oracle that gives conditional independence relations between variables in the observed and interventional distributions. This is clearly a problematic assumption because it implicitly requires unbounded training data. For example, Scheines and Spirtes [SS08] have pointed out that measurement error, quantization and aggregation can easily alter conditional independence relations. The problem of developing finite sample bounds for testing and

learning causal models has been repeatedly posed in the literature. The excellent survey by Guyon, Janzing and Schölkopf [GJS10] on causality from a machine learning perspective underlines the issue as one of the "ten open problems" in the area. To the best of our knowledge, our work is the first to show finite sample complexity and running time bounds for inference problems on causal Bayesian networks.

An application of our learning algorithm is to the problem of *transportability*, studied in [BP13, SP08, LH13, PB11, BP12], which refers to the notion of transferring causal knowledge from a set of source domains to a target domain to identify causal effects in the target domain, when there are certain commonalities between the source and target domains. Most work in this area assume the existence of an algorithm that learns the set of *all* interventions, that is the complete specification of the model, of the source domains. Our learning algorithm can be used for this purpose; it is efficient in terms of time, interventions, and sample complexity, and it learns each intervention distribution to error at most $\epsilon$.

## A.2 Distribution Testing and Learning

There is a vast literature on testing and learning high dimensional distributions in the statistics, and information theory literature, and more recently in computer science with a focus on the computational efficiency of solving such problems. We will not be able to cover and do justice to all of these works in this section. However, we will provide pointers to some of the resources, and also discuss some of the recent progress that is the most closely related to the work we present here.

In the distribution learning and testing framework, the closest to our work is learning and testing graphical models. The seminal work of Chow-Liu [CL68] considered the problem of learning tree-structured graphical models. Motivated by applications across many fields, the problem of learning graphical models from samples has gathered recent interest. Of particular interest is the apparent gap between the sample complexity and computational complexity of learning graphical models. [AKN06, BMS08] provided algorithms for learning bounded degree graphical models with polynomial sample and time complexity. A lower bound on the sample complexity that grows exponentially with the degree, and only logarithmically with the number of dimensions was provided by [SW12], and recent works [Bre15, VMLC16, KM17] have proposed algorithms with near optimal sample complexity, and polynomial running time for learning Ising models.

Sample and computational complexity of testing graphical models has been studied recently, in [CDKS17] for testing Bayesian Networks, and in [DDK18] for testing Ising models. Given sample access to an unknown Bayesian Network, or Ising model, they study the sample complexity, and computation complexity of deciding whether the unknown model is equal to a known fixed model (hypothesis testing).

The problem of testing and learning distribution properties has itself received wide attention in statistics with a history of over a century [Fis25, LR06, CT06]. In these fields, the emphasis is on asymptotic analysis characterizing the convergence rates, and error exponents, as the number of samples tends to infinity. A recent line of work originating from [GR00, BFR+00] focuses on *sublinear* algorithms where the goal is to design algorithms with the number of samples that is smaller than the domain size (e.g., [Can15, Gol17], and references therein).

While most of these results are for learning and testing low dimensional (usually one dimensional) distributions, there are some notable exceptions. Testing for properties such as independence, and monotonicity in high dimensions have been considered recently [BFRV11, ADK15, DK16]. These results show that the optimal sample complexity for testing these properties grows exponentially with the number of dimensions. A line of recent work [DP17, CDKS17, DDK17, DDK18] overcomes this barrier by utilizing additional structure in the high-dimensional distribution induced by Bayesian network or Markov Random Field assumptions.

# B  Proof Sketch for the Fully Observable Case

In the absence of unobservable variables, the analysis becomes much simpler. Let us look at the two-sample testing problem on input causal models $\mathcal{X}$ and $\mathcal{Y}$ defined on a DAG $G$. Now, each c-component is a single vertex, so that every "local" intervention is of the form $P[V_i \mid do(\mathbf{pa}(V_i))]$ for a vertex $V_i$ and an assignment $\mathbf{pa}(V_i)$ to the parents of $V_i$. We define our tester to accept iff each

such local intervention on $\mathcal{X}$ and $\mathcal{Y}$ yields distributions which differ by at most $\epsilon^2/2n$ in squared Hellinger distance. The squared Hellinger distance is defined as follows for two distributions $P$ and $Q$ on $[D]$:

$$H^2(P,Q) := 1 - \sum_{i \in [D]} \sqrt{P(i) \cdot Q(i)} = 1 - BC(P,Q) \tag{4}$$

where $BC(P,Q)$ is the Fidelity or Bhattacharya coefficient of $P$ and $Q$. Below, our subadditivity theorem shows that if the algorithm accepts, then for every intervention, the resulting distributions for $\mathcal{X}$ and $\mathcal{Y}$ differ by at most $\epsilon^2/2$ in squared Hellinger distance, implying $\Delta(\mathcal{X}, \mathcal{Y}) \le \epsilon$.

**Theorem 6.** *Let $\mathcal{X}$ and $\mathcal{Y}$ be two causal Bayesian networks defined on a* known and common *DAG $G$ with no hidden variables. Identify the vertices in $\mathbf{V}$ as $\{V_1, \ldots, V_n\}$ arranged in a topological order. Suppose we know that*

$$H^2(P_{\mathcal{X}}[V_j \mid do(\mathbf{pa}(V_j))], P_{\mathcal{Y}}[V_j \mid do(\mathbf{pa}(V_j))]) \le \gamma \qquad \forall j \in [n], \forall \mathbf{pa}(V_j) \in \Sigma^{|\mathbf{Pa}(V_j)|}. \tag{5}$$

*Then, for each subset $\mathbf{T} \subseteq \mathbf{V}$ and $\mathbf{t} \in \Sigma^{|\mathbf{T}|}$,*

$$H^2\left(P_{\mathcal{X}}[\mathbf{V} \setminus \mathbf{T} \mid do(\mathbf{t})], P_{\mathcal{Y}}[\mathbf{V} \setminus \mathbf{T} \mid do(\mathbf{t})]\right) \le \gamma n. \tag{6}$$

*Proof.* Fix $\mathbf{T} \subseteq \mathbf{V}$ and an assignment $\mathbf{t} \in \Sigma^{|\mathbf{T}|}$. Let $\mathbf{W} = \mathbf{V} \setminus \mathbf{T} = \{W_1, W_2, \ldots, W_m\}$ whose indices are arranged in a topological ordering. By the definition of squared Hellinger distance:

$$H^2\left(\begin{array}{c} P_{\mathcal{X}}[\mathbf{W}|do(\mathbf{t})], \\ P_{\mathcal{Y}}[\mathbf{W}|do(\mathbf{t})] \end{array}\right)$$

$$= 1 - \sum_{w_1, w_2, \ldots, w_m} \sqrt{\frac{P_{\mathcal{X}}[w_1, w_2, \ldots, w_m | do(\mathbf{t})]}{P_{\mathcal{Y}}[w_1, w_2, \ldots, y_m | do(\mathbf{t})]}}$$

$$= 1 - \sum_{w_1, \ldots, w_{m-1}} \sqrt{\frac{P_{\mathcal{X}}[w_1, \ldots, w_{m-1}|do(\mathbf{t})]}{P_{\mathcal{Y}}[w_1, \ldots, w_{m-1}|do(\mathbf{t})]}} \sum_{w_m} \sqrt{\frac{P_{\mathcal{X}}[w_m|w_1, \ldots, w_{m-1}, do(\mathbf{t})]}{P_{\mathcal{Y}}[w_m|w_1, \ldots, w_{m-1}, do(\mathbf{t})]}}$$

$$= 1 - \sum_{w_1, \ldots, w_{m-1}} \sqrt{\frac{P_{\mathcal{X}}[w_1, \ldots, w_{m-1}|do(\mathbf{t})]}{P_{\mathcal{Y}}[w_1, \ldots, w_{m-1}|do(\mathbf{t})]}} \sum_{w_m} \sqrt{\frac{P_{\mathcal{X}}[w_m|do(\mathbf{pa}(W_m))]}{P_{\mathcal{Y}}[w_m|do(\mathbf{pa}(W_m))]}} .$$

The above step can be obtained easily by using Lemma 17 and the conditional independence constraints obtained from $G$. Therefore:

$$H^2\left(\begin{array}{c} P_{\mathcal{X}}[\mathbf{W}|do(\mathbf{t})], \\ P_{\mathcal{Y}}[\mathbf{W}|do(\mathbf{t})] \end{array}\right) \le 1 - \sum_{w_1, \ldots, w_{m-1}} \sqrt{\frac{P_{\mathcal{X}}[w_1, \ldots, w_{m-1}|do(\mathbf{t})]}{P_{\mathcal{Y}}[w_1, \ldots, w_{m-1}|do(\mathbf{t})]}} (1 - \gamma) \quad \text{(from (5))}$$

$$= H^2\left(\begin{array}{c} P_{\mathcal{X}}[W_1 \ldots W_{m-1} \mid do(\mathbf{t})], \\ P_{\mathcal{Y}}[W_1 \ldots W_{m-1}|do(\mathbf{t})] \end{array}\right) (1 - \gamma) + \gamma.$$

By induction on $n$, we get:

$$H^2\left(\begin{array}{c} P_{\mathcal{X}}[\mathbf{W}|do(\mathbf{t})], \\ P_{\mathcal{Y}}[\mathbf{W}|do(\mathbf{t})] \end{array}\right) \le \gamma[1 + (1 - \gamma) + (1 - \gamma)^2 + \ldots + (1 - \gamma)^{m-1}]$$

$$= 1 - (1 - \gamma)^m \le 1 - (1 - \gamma)^n \le n\gamma.$$

$\square$

The time and sample complexities are then determined by that required for two-sample testing on each pair of local distributions with accuracy $\epsilon^2/2n$ in $H^2$ distance. We defer this calculation, as well as bounding the total number of interventions, to later when we analyze semi-Markovian CBNs.

## C  Preliminaries

**Notation.** We use capital (bold capital) letters to denote variables (sets of variables), e.g., $A$ is a variable and $\mathbf{B}$ is a set of variables. We use small (bold small) letters to denote values taken by the corresponding variables (sets of variables), e.g., $a$ is the value of $A$ and $\mathbf{b}$ is the value of the set of variables $\mathbf{B}$. The variables in this paper take values in a discrete set $\Sigma$. We use $[n]$ to denote $\{1, 2, \ldots, n\}$.

**Probability and Statistics.** The total variation (TV) distance between distributions $P$ and $Q$ over the same set $[D]$ is $\delta_{TV}(P,Q) := \frac{1}{2}\sum_{i\in[D]}|P(i) - Q(i)|$. The squared Hellinger distance (given in (4)) and the total variation distance are related by the following.

**Lemma 5** (Hellinger vs total variation)**.** *The Hellinger distance and the total variation distance between two distributions $P$ and $Q$ are related by the following inequality:*

$$H^2(P,Q) \leq \delta_{TV}(P,Q) \leq \sqrt{2H^2(P,Q)}.$$

The problem of two-sample testing for discrete distributions in Hellinger distance, and learning with respect to total variation distance has been studied in the literature, and the following two lemmas state two results we use. Let $P$ and $Q$ denote distributions over a domain of size $D$.

**Lemma 1.** *[Hellinger Test, [DKW18]] Given $O(\min(D^{2/3}/\epsilon^{8/3}, D^{3/4}/\epsilon^2))$ samples from each unknown distributions $P$ and $Q$, we can distinguish between $P = Q$ vs $H^2(P,Q) \geq \epsilon^2$ with probability at least $2/3$. This probability can be boosted to $1 - \delta$ at a cost of an additional $O(\log(1/\delta))$ factor in the sample complexity. The running time of the algorithm is quasi-linear in the sample size* [8].

**Lemma 6** (Learning in TV distance, folklore (e.g. [DL12]))**.** *For all $\delta \in (0,1)$, the empirical distribution $\hat{P}$ computed using $\Theta\left(\frac{D}{\epsilon^2} + \frac{\log\frac{1}{\delta}}{\epsilon^2}\right)$ samples from $P$ satisfies $H^2(P,\hat{P}) \leq \delta_{TV}(P,\hat{P}) \leq \epsilon$, with probability at least $1 - \delta$.*

**Bayesian Networks.** Bayesian networks are popular probabilistic graphical models for describing high-dimensional distributions.

**Definition 2.** *A* Bayesian Network *(BN) $\mathcal{N}$ is a distribution that can be specified by a tuple $\langle \mathbf{V}, G, \{\Pr[V_i \mid \mathbf{pa}(V_i)] : V_i \in \mathbf{V}, \mathbf{pa}(V_i) \in \Sigma^{|\mathbf{Pa}(V_i)|}\}\rangle$ where: (i) $\mathbf{V}$ is a set of variables over alphabet $\Sigma$, (ii) $G$ is a directed acyclic graph with nodes corresponding to the elements of $\mathbf{V}$, and (iii) $\Pr[V_i \mid \mathbf{pa}(V_i)]$ is the conditional distribution of variable $V_i$ given that its parents $\mathbf{Pa}(V_i)$ in $G$ take the values $\mathbf{pa}(V_i)$.*

*The Bayesian Network $\mathcal{N} = \langle \mathbf{V}, G, \{\Pr[V_i \mid \mathbf{pa}(V_i)]\}\rangle$ defines a unique probability distribution $P_\mathcal{N}$ over $\Sigma^{|\mathbf{V}|}$, as follows. For all $\mathbf{v} \in \Sigma^{|\mathbf{V}|}$,*

$$P_\mathcal{N}[\mathbf{v}] = \prod_{V_i \in \mathbf{V}} \Pr[v_i \mid \mathbf{pa}(V_i)].$$

*In this distribution, each variable $V_i$ is independent of its non-descendants given its parents in $G$.*

Conditional independence relations in graphical models are captured by the following definitions.

**Definition 3.** *Given a DAG $G$, a (not necessarily directed) path $p$ in $G$ is said to be* blocked *by a set of nodes $\mathbf{Z}$, if (i) $p$ contains a chain node $B$ ($A \rightarrow B \rightarrow C$) or a fork node $B$ ($A \leftarrow B \rightarrow C$) such that $B \in \mathbf{Z}$ (or) (ii) $p$ contains a collider node $B$ ($A \rightarrow B \leftarrow C$) such that $B \notin \mathbf{Z}$ and no descendant of $B$ is in $\mathbf{Z}$.*

**Definition 4** (d-separation)**.** *For a given DAG $G$ on $\mathbf{V}$, two disjoint sets of vertices $\mathbf{X}, \mathbf{Y} \subseteq \mathbf{V}$ are said to be* d-separated *by $\mathbf{Z}$ in $G$, if every (not necessarily directed) path in $G$ between $\mathbf{X}$ and $\mathbf{Y}$ is blocked by $\mathbf{Z}$.*

**Lemma 7** (Graphical criterion for independence)**.** *For a given BN $\mathcal{N} = \langle \mathbf{V}, G, \{\Pr[V_i \mid \mathbf{pa}(V_i)]\}\rangle$ and $\mathbf{X}, \mathbf{Y}, \mathbf{Z} \subset \mathbf{V}$, if $\mathbf{X}$ and $\mathbf{Y}$ are d-separated by $\mathbf{Z}$ in $G$, then $\mathbf{X}$ is* independent *of $\mathbf{Y}$ given $\mathbf{Z}$ in $P_\mathcal{N}$, denoted by $[\mathbf{X} \perp\!\!\!\perp \mathbf{Y} \mid \mathbf{Z}]$ in $P_\mathcal{N}$.*

## C.1 Causality

We describe Pearl's notion of causality from [Pea95]. Central to his formalism is the notion of an *intervention*. Given a variable set $\mathbf{V}$ and a subset $\mathbf{X} \subset \mathbf{V}$, an intervention $do(\mathbf{x})$ is the process of fixing the set of variables $\mathbf{X}$ to the values $\mathbf{x}$. The *interventional distribution* $\Pr[\mathbf{V} \mid do(\mathbf{x})]$ is the distribution on $\mathbf{V}$ after setting $\mathbf{X}$ to $\mathbf{x}$. As discussed in the introduction, an intervention is quite different from conditioning.

Another important component of Pearl's formalism is that some variables may be unobservable. The unobservable variables can neither be observed nor be intervened. We partition our variable set into two sets $\mathbf{V}$ and $\mathbf{U}$, where the variables in $\mathbf{V}$ are *observable* and the variables in $\mathbf{U}$ are *unobservable*. Given a directed acyclic graph $H$ on $\mathbf{V} \cup \mathbf{U}$ and a subset $\mathbf{X} \subseteq (\mathbf{V} \cup \mathbf{U})$, we use $\mathbf{\Pi}_H(\mathbf{X})$, $\mathbf{Pa}_H(\mathbf{X})$, $\mathbf{An}_H(\mathbf{X})$, and $\mathbf{De}_H(\mathbf{X})$ to denote the set of all parents, observable parents, observable ancestors and observable descendants respectively of $\mathbf{X}$, excluding $\mathbf{X}$, in $H$. When the graph $H$ is clear, we may omit the subscript. As usual, small letters, $\boldsymbol{\pi}(\mathbf{X})$, $\mathbf{pa}(\mathbf{X})$, $\mathbf{an}(\mathbf{X})$ and $\mathbf{de}(\mathbf{X})$ are used to denote their corresponding values. And, we use $H_{\overline{\mathbf{X}}}$ and $H_{\underline{\mathbf{X}}}$ to denote the graph obtained from $H$ by removing the incoming edges to $\mathbf{X}$ and outgoing edges from $\mathbf{X}$ respectively.

**Definition 5** (Causal Bayesian Network). *A causal Bayesian network (CBN) is a collection of interventional distributions that can be defined in terms of a tuple $\langle \mathbf{V}, \mathbf{U}, G, \{\Pr[V_i \mid \boldsymbol{\pi}(V_i)] : V_i \in \mathbf{V}, \boldsymbol{\pi}(V_i) \in \Sigma^{|\mathbf{\Pi}(V_i)|}\}, \{\Pr[U_i \mid \boldsymbol{\pi}(U_i)] : U_i \in \mathbf{U}, \boldsymbol{\pi}(U_i) \in \Sigma^{|\mathbf{\Pi}(U_i)|}\}\rangle$, where (i) $\mathbf{V}$ and $\mathbf{U}$ are the sets of observable and unobservable variables respectively, (ii) $G$ is a directed acyclic graph on $\mathbf{V} \cup \mathbf{U}$, and (iii) $\Pr[V_i \mid \boldsymbol{\pi}(V_i)]$ and $\Pr[U_i \mid \boldsymbol{\pi}(U_i)]$ are the conditional probability distributions of $V_i$ and $U_i$ resp. given that its parents $\mathbf{\Pi}(V_i)$ and $\mathbf{\Pi}(U_i)$ resp. take the values $\boldsymbol{\pi}(V_i)$ and $\boldsymbol{\pi}(U_i))$ resp.*

*A CBN $\mathcal{M} = \langle \mathbf{V}, \mathbf{U}, G, \{\Pr[V_i \mid \boldsymbol{\pi}(V_i)]\}, \{\Pr[U_i \mid \boldsymbol{\pi}(U_i)]\}\rangle$ defines a unique interventional distribution $P_{\mathcal{M}}[\mathbf{V} \mid do(\mathbf{x})]$ for every subset $\mathbf{X} \subseteq \mathbf{V}$ (including $\mathbf{X} = \emptyset$) and assignment $\mathbf{x} \in \Sigma^{|\mathbf{X}|}$, as follows. For all $\mathbf{v} \in \Sigma^{|\mathbf{V}|}$:*

$$P_{\mathcal{M}}[\mathbf{v} \mid do(\mathbf{x})] = \begin{cases} \sum_{\mathbf{u}} \prod_{V_i \in \mathbf{V} \setminus \mathbf{X}} \Pr[v_i \mid \boldsymbol{\pi}(V_i)] \cdot \prod_{U_i \in \mathbf{U}} \Pr[u_i \mid \boldsymbol{\pi}(U_i)] & \text{if } \mathbf{v} \text{ is consistent with } \mathbf{x} \\ 0 & \text{otherwise.} \end{cases}$$

*We say that $G$ is the* causal graph *corresponding to the CBN $\mathcal{M}$.*

Another equivalent way to define a CBN is by specifying the set of interventional distributions $P_{\mathcal{M}}[\mathbf{V} \mid do(\mathbf{x})]$ for all subsets $\mathbf{X}$ and assignments $\mathbf{x}$. To connect to the preceding definition, we require that each $P_{\mathcal{M}}[\mathbf{V} \mid do(\mathbf{x})]$ is defined by the Bayesian network described by $G_{\overline{\mathbf{X}}}$ with the conditional probability distributions obtained by setting the variables in $\mathbf{X}$ to the constants $\mathbf{x}$.

It is standard in the causality literature to work with causal graphs of a particular structure:

**Definition 6** (Semi-Markovian causal graph and Semi-Markovian Bayesian network). *A semi-Markovian causal graph (SMCG) $G$ is a directed acyclic graph on $\mathbf{V} \cup \mathbf{U}$ where every unobservable variable is a root node and has exactly two children, both observable. A semi-Markovian Bayesian network (SMBN) is a causal Bayesian network where the causal graph is semi-Markovian.*

There exists a known reduction (described formally in Appendix I) from general causal Bayesian networks to semi-Markovian Bayesian networks that preserves all the properties we use in our analysis, so that henceforth, we will restrict only to SMBNs.

In SMCGs, the divergent edges $V_i \leftarrow U_k \rightarrow V_j$ are usually represented by *bi-directed edges* $V_i \leftrightarrow V_j$. A bi-directed edge between two observable variables implicitly represents the presence of an unobservable parent.

**Definition 7** (c-component). *For a given SMCG $G$, $\mathbf{S} \subseteq \mathbf{V}$ is a c-component of $G$, if $\mathbf{S}$ is a maximal set such that between any two vertices of $\mathbf{S}$, there exists a path that uses only bi-directed edges.*

Since a c-component forms an equivalence relation, the set of all c-components forms a partition of $\mathbf{V}$, the observable vertices of $G$. We use the notation $C(\mathbf{V}) = \{\mathbf{S}_1, \mathbf{S}_2, \ldots, \mathbf{S}_k\}$ to denote the partition of $\mathbf{V}$ into the c-components of $G$, where each $\mathbf{S}_i \subseteq \mathbf{V}$ is a c-component of $G$.

Also, for $\mathbf{X} \subseteq \mathbf{V}$, the induced subgraph $G[\mathbf{X}]$ is the subgraph obtained by removing the vertices $\mathbf{V} \setminus \mathbf{X}$ and their corresponding edges from $G$. We use the notation $C(\mathbf{X}) = \{\mathbf{S}_1, \mathbf{S}_2, \ldots, \mathbf{S}_k\}$ to denote the set of all c-components of $G[\mathbf{X}]$, that is each $\mathbf{S}_i \subseteq \mathbf{X}$ is a c-component of $G[\mathbf{X}]$. The next two lemmas capture the factorizations of distributions in SMBN.

**Lemma 8.** *Let $\mathcal{M}$ be a given SMBN with respect to the SMCG $G$. For any set $\mathbf{S} \subseteq \mathbf{V}$, and a subset $\mathbf{D}$ such that $(\mathbf{V} \setminus \mathbf{S}) \supseteq \mathbf{D} \supseteq \mathbf{Pa}(\mathbf{S})$, and for any assignment $\mathbf{s}, \mathbf{d}$, $P_{\mathcal{M}}[\mathbf{s} \mid do(\mathbf{d})] = P_{\mathcal{M}}[\mathbf{s} \mid do(\mathbf{pa}(\mathbf{S}))]$, where $\mathbf{pa}(\mathbf{S})$ denotes the assignment consistent with $\mathbf{d}$.*

*Proof.* When the parents of $\mathbf{S}$, $\mathbf{Pa}(\mathbf{S})$, are targeted for intervention, the distribution on $\mathbf{S}$ remains the same irrespective of whether the other vertices in $(\mathbf{V} \setminus \mathbf{S})$ are intervened or not. $\qquad\square$

**Lemma 9** (c-component factorization, [TP02]). *Given a SMBN $\mathcal{M}$ with respect to the causal graph $G$ and a subset $\mathbf{X} \subseteq \mathbf{V}$, let $C(\mathbf{V} \backslash \mathbf{X}) = \{\mathbf{S}_1, \ldots, \mathbf{S}_k\}$. For any given assignment $\mathbf{v}$,*

$$P_{\mathcal{M}}[\mathbf{v} \backslash \mathbf{x} \mid do(\mathbf{x})] = \prod_i P_{\mathcal{M}}[\mathbf{s}_i \mid do(\mathbf{v} \backslash \mathbf{s}_i)].$$

For a given SMCG $G$, the in-degree and out-degree of an observable vertex $V_i \in \mathbf{V}$ denote the number of observable parents and observable children of $V_i$ in $G$ respectively. The maximum in-degree of a SMCG $G$ is the maximum in-degree over all the observable vertices. The maximum degree of a SMCG $G$ is the maximum of the sum of the in-degree and out-degree over all the observable vertices.

**Definition 8** (Graphs with *bounded in-degree* and *bounded c-component*). $\mathcal{G}_{d,\ell}$ *denotes the class of* SMCG*s with maximum in-degree at most $d$ and the size of the largest c-component at most $\ell$.*

### C.2 Problem Definitions

Here we define the testing and learning problems considered in the paper. Let $\mathcal{M}$ and $\mathcal{N}$ be two SMBNs. We say that $\mathcal{M} = \mathcal{N}$, if

$$P_{\mathcal{M}}[\mathbf{V} \backslash \mathbf{T} \mid do(\mathbf{t})] = P_{\mathcal{N}}[\mathbf{V} \backslash \mathbf{T} \mid do(\mathbf{t})] \qquad \forall \mathbf{T} \subseteq \mathbf{V}, \mathbf{t} \in \Sigma^{|\mathbf{T}|}.$$

And we say that $\Delta(\mathcal{M}, \mathcal{N}) > \epsilon$, if there exists $\mathbf{T} \subseteq \mathbf{V}$ and $\mathbf{t} \in \Sigma^{|\mathbf{T}|}$ such that

$$\delta_{TV}(P_{\mathcal{M}}[\mathbf{V} \backslash \mathbf{T} \mid do(\mathbf{t})], P_{\mathcal{N}}[\mathbf{V} \backslash \mathbf{T} \mid do(\mathbf{t})]) > \epsilon.$$

**Definition 9** (Causal Goodness-of-fit Testing (CGFT$(G, \mathcal{M}, \epsilon)$)). *Given a SMCG $G$, a (known) SMBN $\mathcal{M}$ on $G$, and $\epsilon > 0$. Let $\mathcal{X}$ denote an unknown SMBN on $G$. The objective of CGFT$(G, \mathcal{M}, \epsilon)$ is to distinguish between $\mathcal{X} = \mathcal{M}$ versus $\Delta(\mathcal{X}, \mathcal{M}) > \epsilon$ with probability at least 2/3, by performing interventions and taking samples from the resulting interventional distributions of $\mathcal{X}$.*

**Definition 10** (Causal Two-sample Testing (C2ST$(G, \epsilon)$)). *Given a SMCG $G$, and $\epsilon > 0$. Let $\mathcal{X}$ and $\mathcal{Y}$ be two unknown SMBNs on $G$. The objective of C2ST$(G, \epsilon)$ is to distinguish between $\mathcal{X} = \mathcal{Y}$ versus $\Delta(\mathcal{X}, \mathcal{Y}) > \epsilon$ with probability at least 2/3, by performing interventions and taking samples from the resulting interventional distributions of $\mathcal{X}$ and $\mathcal{Y}$.*

**Definition 11** (Learning SMBNs (CL$(G, \epsilon)$)). *Given a SMCG $G$ and $\epsilon > 0$. Let $\mathcal{X}$ be an unknown SMBN on $G$. The objective of CL$(G, \epsilon)$ is to perform interventions and taking samples from the resulting interventional distributions of $\mathcal{X}$, and return an oracle that for any $\mathbf{T} \subseteq \mathbf{V}$ and $\mathbf{t} \in \Sigma^{|\mathbf{T}|}$ returns an estimated interventional distribution $P_{ES}[\mathbf{V} \backslash \mathbf{T} \mid do(\mathbf{t})]$ such that*

$$\delta_{TV}([P_{\mathcal{X}}[\mathbf{V} \backslash \mathbf{T} \mid do(\mathbf{t})], P_{ES}[\mathbf{V} \backslash \mathbf{T} \mid do(\mathbf{t})]) < \epsilon.$$

We emphasize that in all three problems, the causal graph $G$ is known explicitly in advance.

## D Testing and Learning Algorithms for SMBNs

### D.1 Testing

Recall that in Section 2, we provided an algorithm for the two-sample testing problem when the SMCG $G$ is common and known. We will now consider the problem of two sample testing, where $\mathcal{X}$ and $\mathcal{Y}$ are still on the same common SMCG $G$, but $G$ is unknown. We now show an algorithm that uses the same number of interventions and samples as Theorem 4 for the known $G$ case, however requiring $O(n^{\ell+1} K^{\ell(2d+7/4)} \epsilon^{-2})$ time.

**Theorem 7** (Algorithm for C2ST$(G, \epsilon)$ – Unknown graph). *Consider the same set-up as Theorem 4, except that the SMCG $G \in \mathcal{G}_{d,\ell}$ is unknown. Then, there is an algorithm to this problem, that makes $O(K^{\ell d}(3d)^{\ell} \log n)$ interventions to $\mathcal{X}$ and $\mathcal{Y}$, taking $O(K^{\ell(d+7/4)} n\epsilon^{-2})$ samples per intervention, in time $\tilde{O}(n^{\ell} K^{\ell(2d+7/4)} n\epsilon^{-2})$.*

*Proof.* We first use Lemma 10 and obtain a set of interventions $\mathbf{I}$, such that $\mathbf{I}$ is a covering set with error probability at most $1/6$. Note that Lemma 10 holds even when the underlying graph $G$ is unknown.

---

**Algorithm 2**: Algorithm for $\mathsf{C2ST}(G, \epsilon)$ – Unknown graph

**I**: Covering intervention set

1. Under each intervention $I = \Pr[\mathbf{V} \setminus \mathbf{T} \mid do(\mathbf{t})] \in \mathbf{I}$:

   (a) Obtain $O(K^{\ell(d+7/4)} n \epsilon^{-2})$ samples from the interventional distribution of $I$ in both models $\mathcal{X}$ and $\mathcal{Y}$.

   (b) For each subset $\mathbf{S} \subseteq \mathbf{V} \setminus \mathbf{T}$ of size $\leq \ell$, using Lemma 1, Lemma 8 and the obtained samples, test (with error probability at most $1/(6K^{\ell d} 2^{\ell} n)$):

   $$P_{\mathcal{X}}[\mathbf{S} \mid do(\mathbf{t})] = P_{\mathcal{Y}}[\mathbf{S} \mid do(\mathbf{t})] \text{ versus } H^2 \left( \begin{array}{c} P_{\mathcal{X}}[\mathbf{S} \mid do(\mathbf{t})], \\ P_{\mathcal{Y}}[\mathbf{S} \mid do(\mathbf{t})] \end{array} \right) \geq \frac{\epsilon^2}{2K^{\ell(d+1)} n}$$

   Output "$\Delta(\mathcal{X}, \mathcal{Y}) > \epsilon$" if the latter.

2. Output "$\mathcal{X} = \mathcal{Y}$".

---

For each intervention, we go over all subsets $S$ of size $\leq \ell$. Therefore we perform at most $\binom{n}{\leq \ell} = O(n^{\ell})$ sub-tests for an intervention. For each sub-test, the algorithm's running time is quasi-linear in the sample complexity (Lemma 1), therefore taking a total time of $O(n^{\ell} K^{\ell(2d+7/4)} n \epsilon^{-2})$. The number of interventions follow from Lemma 10 and the number of samples follow from the algorithm.

**Correctness.** As in the proof of Theorem 4, we use Theorem 3 to show that when $\Delta(\mathcal{X}, \mathcal{Y}) > \epsilon$, then there exists a subset $\mathbf{S}$ of some c-component and an $I \in \mathbf{I}$ that does not intervene any node in $\mathbf{S}$ but intervenes $\mathbf{Pa}(\mathbf{S})$ with some assignment $\mathbf{pa}(\mathbf{s})$ such that

$$H^2(P_{\mathcal{X}}[\mathbf{S} \mid do(\mathbf{pa}(\mathbf{S}))], P_{\mathcal{Y}}[\mathbf{S} \mid do(\mathbf{pa}(\mathbf{S}))]) > \epsilon^2/(2K^{\ell(d+1)} n).$$

This together with Lemma 3 proves that $P_{\mathcal{X}}$ and $P_{\mathcal{Y}}$ are far in terms of the total variation distance. Since the error probability of each sub-test is bounded by at most $1/(6K^{\ell d} 2^{\ell} n)$ and the error probability of **I** being a covering intervention set is at most $1/6$, by union bound, we will have an error of at most $1/3$ over the entire algorithm. $\square$

## D.2 Learning

Our next result is on learning SMBNs over a known causal graph. Our algorithm is improper, meaning that it does not output a causal model in the form of an SMBN, but rather outputs an oracle which succinctly encodes all the interventional distributions. See Definition 11 for a rigorous formulation of the problem.

**Theorem 8** (Algorithm for $\mathsf{CL}(G, \epsilon)$)**.** *For any given SMCG $G \in \mathcal{G}_{d,\ell}$ with $n$ vertices and a parameter $\epsilon > 0$, there exists an algorithm that takes as input an unknown SMBN $\mathcal{X}$ over $G$, that performs $O(K^{\ell d}(3d)^{\ell} \log n)$ interventions to $\mathcal{X}$, taking $\tilde{O}(K^{\ell(2d+3)} n^2 \epsilon^{-4})$ samples per intervention, that runs in time $\tilde{O}\left(2^{\ell} K^{\ell(3d+3)} n^3 \epsilon^{-4}\right)$, and that with probability at least $2/3$, outputs an oracle $\mathcal{N}$ with the following behavior. Given as input any $\mathbf{T} \subseteq \mathbf{V}$ and assignment $\mathbf{t} \in \Sigma^{|\mathbf{T}|}$, $\mathcal{N}$ outputs an interventional distribution $P_{\mathcal{N}}[\mathbf{V} \setminus \mathbf{T} | do(\mathbf{t})]$ such that:*

$$\delta_{TV}(P_{\mathcal{X}}[\mathbf{V} \setminus \mathbf{T} \mid do(\mathbf{t})], P_{\mathcal{N}}[\mathbf{V} \setminus \mathbf{T} \mid do(\mathbf{t})]) < \epsilon$$

*When the maximum degree (in-degree plus out-degree) of $G$ is bounded by $d$, then our algorithm uses $O(K^{\ell d}(3d)^{\ell} \ell d^2 \log K)$ interventions with the same sample complexity and running time as above.*

---

**Algorithm 3**: Algorithm for $\mathsf{CL}(G, \epsilon)$

---

**I**: Covering intervention set

1. Under each intervention $I \in \mathbf{I}$:

   (a) Obtain $\tilde{O}(n^2 K^{\ell(2d+3)} \epsilon^{-4})$ samples from the interventional distribution of $I$ in $\mathcal{X}$.

   (b) For each subset $\mathbf{S}$ of a c-component, if $I$ does not set $\mathbf{S}$ but sets $\mathbf{Pa(S)}$ to $\mathbf{pa(S)}$, use Lemma 6, Lemma 8 and the obtained samples to learn:

   $$P_{\mathcal{N}}[\mathbf{S}|do(\mathbf{pa(S)})] \text{ s.t., } H^2(P_{\mathcal{N}}[\mathbf{S}|do(\mathbf{pa(S)})], P_{\mathcal{X}}[\mathbf{S}|do(\mathbf{pa(S)})]) \leq \frac{\epsilon^2}{2K^{\ell(d+1)}n}$$

   with probability of error at most $1/(3K^{\ell d}2^{\ell}n)$.

2. Return the following oracle $\mathcal{N}$ that takes as input: $\mathbf{T} \subseteq \mathbf{V}$ and $\mathbf{t} \in \Sigma^{|\mathbf{T}|}$

   (i) Let $C(\mathbf{V} \setminus \mathbf{T}) = \{\mathbf{S}_1, \ldots, \mathbf{S}_p\}$.

   (ii) Output the distribution $P_{\mathcal{N}}[\mathbf{V} \setminus \mathbf{T} \mid do(\mathbf{t})]$ where for any assignment $\mathbf{v} \setminus \mathbf{t}$:

   $$P_{\mathcal{N}}[\mathbf{v} \setminus \mathbf{t} \mid do(\mathbf{t})] = \prod_{i=1}^{p} P_{\mathcal{N}}[\mathbf{s}_i \mid do(\mathbf{v} \setminus \mathbf{s}_i)]$$

---

The covering intervention set used in the algorithm above is as defined in Definition 1.

**Number of interventions, time, and sample requirements.** The number of interventions is obtained using the bound on the size of the covering intervention set from Lemma 10. When the maximum degree is bounded, we can use Lemma 11. The number of samples per intervention is obtained from Lemma 6. Since the algorithm learns at most $nK^{\ell d}2^{\ell}$ interventions (subroutines), and each subroutine takes time linear in the sample size, the time complexity follows.

**Correctness.** For any given $\mathbf{T}$, $do(\mathbf{t})$, let $C(\mathbf{V} \setminus \mathbf{T}) = \{\mathbf{S}_1, \ldots, \mathbf{S}_p\}$. Lemma 9 justifies that

$$P_{\mathcal{N}}[\mathbf{v} \setminus \mathbf{t} \mid do(\mathbf{t})] = \prod_{i} P_{\mathcal{N}}[\mathbf{s}_i \mid do(\mathbf{v} \setminus \mathbf{s}_i)].$$

Similar to the proof of Theorem 4, using Theorem 3 and Lemma 5, we get:

$$H^2(P_{\mathcal{N}}[\mathbf{V} \setminus \mathbf{T} \mid do(\mathbf{t})], P_{\mathcal{X}}[\mathbf{V} \setminus \mathbf{T} \mid do(\mathbf{t})]) < \epsilon^2/2$$
$$\implies \delta_{TV}(P_{\mathcal{N}}[\mathbf{V} \setminus \mathbf{T} \mid do(\mathbf{t})], P_{\mathcal{X}}[\mathbf{V} \setminus \mathbf{T} \mid do(\mathbf{t})]) < \epsilon.$$

# E  Main Ingredients of the Analysis

## E.1  Covering Intervention Sets

Let $G \in \mathcal{G}_{d,\ell}$. $G$ contains (i) at most $n$ c-components; (ii) for each c-component, there are at most $2^{\ell}$ possible subsets; and (iii) for every subset of a c-component, there can be at most $\ell d$ observable parents. Hence, the trivial bound on the number of interventions required by our algorithms (Algorithm 1,2 and 3), i.e., covering set of interventions, is $n2^{\ell}\Sigma^{\ell d}$. However, the size of the covering set can be further improved. For example, suppose we have a collection of $n/(d+1)$ disjoint stars, where each star has $d$ arms (directed inwards). Then, there is a covering set of interventions of size $O((|\Sigma|+1)^d)$. The reason is that the same intervention can be applied to each star in parallel, and the number of interventions to each star is bounded by a function of $d$ and $|\Sigma|$ and does not depend on $n$. A generalization of this argument for any $G \in \mathcal{G}_{d,\ell}$ provides a covering set of interventions of size $O(K^{\ell d}(3d)^{\ell}(\log n + \ell d \log K))$.

**Lemma 10** (Counting Lemma: bounded in-degree). *Let $G \in \mathcal{G}_{d,\ell}$ be a SMCG with $n$ vertices and $\Sigma$ be an alphabet set of size $K$. Then, there is a randomized algorithm that outputs a set $\mathbf{I}$ of size*

$O(K^{\ell d}(3d)^\ell(\log n + \ell d \log K + \log(1/\delta)))$. *such that, with probability at least $1 - \delta$, $\mathbf{I}$ is a covering intervention set.*

*Proof.* Let $t = K^{\ell d}(3d)^\ell(\log n + 2\ell d \log K + \log(1/\delta))$. The interventions in $\mathbf{I}$ are chosen by the following procedure: For each $j \in [t]$ and for each $V_i \in V$, $V_i$ is observed in $I_j$ with probability $1/(d+1)$ and otherwise, $V_i$ is intervened with the assignment chosen uniformly from $\Sigma$. Let $V_i = *$ denotes that $V_i$ is not intervened. Consider a fixed c-component $\mathbf{C}$, a fixed subset $\mathbf{S} \subseteq \mathbf{C}$, a fixed assignment $\mathbf{pa}(\mathbf{S}) \in \Sigma^{|\mathbf{Pa}(\mathbf{S})|}$ and a fixed $j \in [t]$. Now,

$$
\begin{aligned}
\Pr[I_j(\mathbf{S}) = *^{|\mathbf{S}|} \wedge I_j(\mathbf{Pa}(\mathbf{S})) = \mathbf{pa}(\mathbf{S})] &= \left(\frac{1}{d+1}\right)^{|\mathbf{S}|} \cdot \left(\frac{d}{K(d+1)}\right)^{|\mathbf{Pa}(\mathbf{S})|} \\
&\geq (d+1)^{-\ell} K^{-\ell d} e^{-\ell} \text{ [Since } |\mathbf{Pa}(\mathbf{S})| \leq \ell d \text{ and } |\mathbf{S}| \leq \ell] \\
&\geq (3d)^{-\ell} K^{-\ell d}.
\end{aligned}
$$

This implies that

$$
\Pr[\forall j \in [t], (I_j(\mathbf{S}) \neq *^{|\mathbf{S}|} \vee I_j(\mathbf{Pa}(\mathbf{S})) \neq \mathbf{pa}(\mathbf{S}))] \leq \left(1 - (3d)^{-\ell} K^{-\ell d}\right)^t \leq \frac{\delta}{n} K^{-2\ell d}.
$$

Hence,

$$
\Pr[\exists \mathbf{C}, \exists \mathbf{S} \subseteq \mathbf{C}, \exists \mathbf{pa}(\mathbf{S}) \in \Sigma^{|\mathbf{Pa}(\mathbf{S})|}, \forall j \in [t], (I_j(\mathbf{S}) \neq *^{|\mathbf{S}|} \vee I_j(\mathbf{Pa}(\mathbf{S})) \neq \mathbf{pa}(\mathbf{S}))]
$$
$$
\leq n 2^\ell K^{\ell d} \cdot \frac{\delta}{n} K^{-2\ell d} \leq \delta
$$

by the union bound. $\qquad\square$

**Remark 1.** *The above proof can be made deterministic by using explicit deterministic constructions of almost $\ell d$-wise independent random variables [AGHP92, EGL+92, NN90].*

To illustrate Remark 1, we will require the following definition.

**Definition 12** (almost $k$-wise independence). *Let $\mathcal{S} \subset \{0,1\}^m$ be a sample space and let $\mathbf{X} = \{X_1, X_2, \ldots, X_m\}$ be chosen uniformly from $\mathcal{S}$. $\mathcal{S}$ is $(\epsilon, k)$-independent if for any $k$ positions $i_1 < i_2 < \cdots < i_k$ and for any $k$ bit string $\boldsymbol{\alpha} \in \{0,1\}^k$,*

$$
|\Pr[X_{i_1}, X_{i_2}, \ldots, X_{i_k} = \boldsymbol{\alpha}] - 2^{-k}| \leq \epsilon.
$$

[NN90] presented an efficient construction of a sample space $\mathcal{A} \subset \{0,1\}^m$ of size $O(k \log m \cdot 2^{2k} \cdot \frac{1}{\epsilon^4})$ such that $\mathcal{A}$ is $(\epsilon, k)$-independent.

Let $|\Sigma|$ be an alphabet set of size $K$, $m = n \log K$ and $k = \ell d \log K$. Let $\mathcal{S} \subseteq \{0,1\}^m$ represent the binary encoding of the set of all possible assignments of the observable vertices $\mathbf{V}$. To obtain Remark 1, it is sufficient to construct a *small* set of assignments $\mathcal{A} \subset \{0,1\}^m$ such that: for every $\boldsymbol{\alpha} \in \{0,1\}^k$ and for any $k$ indices $i_1 < i_2 < \cdots < i_k$, there exists an assignment $\mathbf{a} = (a_1, \ldots, a_m) \in \mathcal{A}$ such that $a_{i_1}, a_{i_2}, \ldots, a_{i_k} = \boldsymbol{\alpha}$. Also note that any sample space $\mathcal{A} \subset \{0,1\}^m$ which is $(2^{-k-1}, k)$-independent will achieve the desired property. Hence, the construction of [NN90] yields the required set of assignments $\mathcal{A}$ of size $O(\ell d \cdot \log K \cdot K^{6\ell d} \cdot \log n)$.

For bounded-degree graphs, we can use the Lovász local lemma and further minimize the size of the covering set.

**Lemma 11.** *[Counting Lemma: bounded total degree] Let $G \in \mathcal{G}_{d,\ell}$ be an SMCG with $n$ vertices, whose variables take values in $\Sigma$ with $|\Sigma| = K$, and whose maximum degree is bounded by $d$. Then, there exists covering intervention set $\mathbf{I}$ of size $O(K^{\ell d}(3d)^\ell \ell d^2 \log K)$.*

*Proof.* Let $t = K^{\ell d}(3d)^\ell(\ell d^2 + \ell d \log K + 2)$. The interventions in $\mathbf{I}$ are chosen by the following procedure: For each $j \in [t]$ and for each $V_i \in V$, $V_i$ is observed in $I_j$ with probability $1/(d+1)$ and otherwise, $V_i$ is intervened with the assignment chosen uniformly from the set $\Sigma$. Let $V_i = *$ denotes that $V_i$ is observed (not intervened).

For a fixed set $\mathbf{S}$ that is a subset of a c-component and a fixed assignment $\mathbf{pa}(\mathbf{S}) \in \Sigma^{|\,\mathbf{Pa}(\mathbf{S})|}$, let $A_{\mathbf{S},\mathbf{pa}(\mathbf{S})}$ be the event: $\forall j \in [t], (I_j(\mathbf{S}) \neq *^{|\mathbf{S}|} \vee I_j(\mathbf{Pa}(\mathbf{S})) \neq \mathbf{pa}(\mathbf{S}))$. Similar to the proof of Lemma 10, for any fixed $\mathbf{S}$ and $\mathbf{pa}(\mathbf{S})$: $\Pr[A_{\mathbf{S},\mathbf{pa}(\mathbf{S})}] \leq 1/(42^{\ell d^2} K^{\ell d})$.

Now, note that $A_{\mathbf{S},\mathbf{pa}(\mathbf{S})}$ and $A_{\mathbf{T},\mathbf{pa}(\mathbf{T})}$ are independent if $\mathbf{Pa}(\mathbf{S})$ and $\mathbf{Pa}(\mathbf{T})$ are disjoint. For a fixed $\mathbf{S}$, the number of subsets $\mathbf{T}$ such that $\mathbf{Pa}(\mathbf{S}) \cap \mathbf{Pa}(\mathbf{T}) \neq \emptyset$ is at most $2^{\ell d^2}$ (since, the number of children of the parents of $S$ is at most $\ell d^2$). Therefore, for a fixed $\mathbf{S}$ and $\mathbf{pa}(\mathbf{S})$, $A_{\mathbf{S},\mathbf{pa}(\mathbf{S})}$ is independent of all $A_{\mathbf{T},\mathbf{pa}(\mathbf{T})}$'s except for at most $2^{\ell d^2} K^{\ell d}$ many of them (taking into account the number of possible assignments $\mathbf{pa}(\mathbf{T})$). Hence, the Lovász Local Lemma [AS04, Chapter 5] guarantees that there exists a set of $t$ interventions such that $\neg A_{\mathbf{S},\mathbf{pa}(\mathbf{S})}$ for all $\mathbf{S}$ and $\mathbf{pa}(\mathbf{S})$. $\square$

**Remark 2** (Explicitness)**.** *Although Lemma 11 only asserts the existence of a covering intervention, its proof can be turned into a linear time algorithm using the constructive proofs of the Lovász Local Lemma [Mos09, MT10].*

## E.2 Subadditivity Theorem for SMBNs

The next theorem states that if two causal models are "far", then they must be "far" under some "local" intervention.

**Theorem 3.** (Subadditivity Theorem) *Let $\mathcal{M}$ and $\mathcal{N}$ be two* SMBN*s defined on a* known and common SMCG $G \in \mathcal{G}_{d,\ell}$. *For a given intervention $do(\mathbf{t})$, let $\mathbf{V} \setminus \mathbf{T}$ partition into $\mathcal{C} = \{\mathbf{C}_1, \mathbf{C}_2, \ldots, \mathbf{C}_p\}$, the c-components with respect to the induced graph $G[\mathbf{V} \setminus \mathbf{T}]$. Suppose*

$$H^2(P_{\mathcal{M}}[\mathbf{C}_j \mid do(\mathbf{pa}(\mathbf{C}_j))], P_{\mathcal{N}}[\mathbf{C}_j \mid do(\mathbf{pa}(\mathbf{C}_j))]) \leq \gamma \qquad \forall j \in [p], \forall \mathbf{pa}(\mathbf{C}_j) \in \Sigma^{|\,\mathbf{Pa}(\mathbf{C}_j)|}. \tag{2}$$

*Then*

$$H^2(P_{\mathcal{M}}[\mathbf{V} \setminus \mathbf{T} \mid do(\mathbf{t})], P_{\mathcal{N}}[\mathbf{V} \setminus \mathbf{T} \mid do(\mathbf{t})]) \leq \epsilon \qquad \forall \mathbf{t} \in \Sigma^{|\mathbf{T}|} \tag{3}$$

*where $\epsilon = \gamma |\Sigma|^{\ell(d+1)} n$.*

*Proof.* Let $\mathbf{W} = \mathbf{V} \setminus \mathbf{T} = \{W_1, \ldots, W_r\}$, where the indices are arranged in a topological ordering. Here we focus only on distributions on $\mathbf{W}$ after the intervention $do(\mathbf{t})$. That is, our focus is restricted to the graph $G_{\overline{\mathbf{T}}}$, the intervention $do(\mathbf{t})$ and the vertices $\mathbf{W} = \mathbf{V} \setminus \mathbf{T}$. We know that

$$H^2(P_{\mathcal{M}}[\mathbf{W} \mid do(\mathbf{t})], P_{\mathcal{N}}[\mathbf{W} \mid do(\mathbf{t})]) = 1 - \sum_{\mathbf{w}} \sqrt{P_{\mathcal{M}}[\mathbf{w} \mid do(\mathbf{t})] P_{\mathcal{N}}[\mathbf{w} \mid do(\mathbf{t})]}$$

$$= 1 - BC(P_{\mathcal{M}}[\mathbf{W} \mid do(\mathbf{t})], P_{\mathcal{N}}[\mathbf{W} \mid do(\mathbf{t})]) \tag{7}$$

where $BC(P_{\mathcal{M}}[\mathbf{W} \mid do(\mathbf{t})], P_{\mathcal{N}}[\mathbf{W} \mid do(\mathbf{t})])$ is the Bhattacharya coefficient of $P_{\mathcal{M}}[\mathbf{W} \mid do(\mathbf{t})]$ and $P_{\mathcal{N}}[\mathbf{W} \mid do(\mathbf{t})]$ (see (4)).

For each $j \in [p]$, identify the vertices in $\mathbf{C}_j$ as $\{W_{n_{j,1}}, \ldots, W_{n_{j,s_j}}\}$ where $s_j = |\mathbf{C}_j|$ and $n_{j,1} < \cdots < n_{j,s_j}$. Using Lemma 9, we express the distributions in terms of the product $\prod_{j=1}^{p} \Pr[\mathbf{c}_j \mid do(\mathbf{w} \setminus \mathbf{c}_j)]$ [TP02],

$BC(P_{\mathcal{M}}[\mathbf{W} \mid do(\mathbf{t})], P_{\mathcal{N}}[\mathbf{W} \mid do(\mathbf{t})])$

$$= \sum_{\mathbf{w}} \sqrt{\frac{P_{\mathcal{M}}[\mathbf{w} \mid do(\mathbf{t})]}{P_{\mathcal{N}}[\mathbf{w} \mid do(\mathbf{t})]}}$$

$$= \sum_{\mathbf{w}} \sqrt{\prod_{j=1}^{p} \frac{P_{\mathcal{M}}[\mathbf{c}_j \mid do(\mathbf{w} \setminus \mathbf{c}_j)]}{P_{\mathcal{N}}[\mathbf{c}_j \mid do(\mathbf{w} \setminus \mathbf{c}_j)]}}$$

$$= \sum_{\mathbf{w}} \sqrt{\prod_{j=1}^{p} \prod_{i=1}^{s_j} \frac{P_{\mathcal{M}}[w_{n_{j,i}} \mid w_{n_{j,1}}, \ldots, w_{n_{j,i-1}}, do(\mathbf{w} \setminus \mathbf{c}_j)]}{P_{\mathcal{N}}[w_{n_{j,i}} \mid w_{n_{j,1}}, \ldots, w_{n_{j,i-1}}, do(\mathbf{w} \setminus \mathbf{c}_j)]}}$$

$$= \sum_{\mathbf{w}} \sqrt{\prod_{j=1}^{p} \prod_{i=1}^{s_j} \frac{P_{\mathcal{M}}[w_{n_{j,i}} \mid w_{n_{j,1}}, \ldots, w_{n_{j,i-1}}, do(\mathbf{pa}(W_{n_{j,1}}, \ldots, W_{n_{j,i-1}}))]}{P_{\mathcal{N}}[w_{n_{j,i}} \mid w_{n_{j,1}}, \ldots, w_{n_{j,i-1}}, do(\mathbf{pa}(W_{n_{j,1}}, \ldots, W_{n_{j,i-1}}))]}} \quad \text{(using Lemma 17).}$$

$$\tag{8}$$

The column-wise notation used above (within the square root) represent the multiplication of all those terms inside the square root, and is only used to represent lengthy multiplications in a single line.

For $i \in [s_j]$, let

$$dep(n_{j,i}) := \{W_{n_{j,1}}, \ldots, W_{n_{j,i}}\} \cup (\mathbf{Pa}(\{W_{n_{j,1}}, \ldots, W_{n_{j,i}}\}) \setminus \mathbf{T}).$$

For $j \in [p], i \in [s_j]$, let $X_{n_{j,i}} : \Sigma^{|dep(n_{j,i})|} \to [0,1]$ be

$$X_{n_{j,i}}(\mathbf{w}_{dep(n_{j,i})}) := \sqrt{\frac{P_{\mathcal{M}}[w_{n_{j,i}} \mid w_{n_{j,1}}, \ldots, w_{n_{j,i-1}}, do(\mathbf{pa}(W_{n_{j,1}}, \ldots, W_{n_{j,i-1}}))]}{P_{\mathcal{N}}[w_{n_{j,i}} \mid w_{n_{j,1}}, \ldots, w_{n_{j,i-1}}, do(\mathbf{pa}(W_{n_{j,1}}, \ldots, W_{n_{j,i-1}}))]}} \quad .$$

Recall that indices of $\mathbf{W}$ follow a topological ordering. Using this topological ordering and plugging in the expression above, we obtain

$$BC(P_{\mathcal{M}}[\mathbf{W} \mid do(\mathbf{t})], P_{\mathcal{N}}[\mathbf{W} \mid do(\mathbf{t})]) = \sum_{w_1} X_1(\mathbf{w}_{dep(1)}) \sum_{w_2} X_2(\mathbf{w}_{dep(2)}) \ldots \sum_{w_r} X_r(\mathbf{w}_{dep(r)}) \tag{9}$$

where $r = |\mathbf{W}|$. In order to prove the theorem, it will suffice to prove that this expression is at least $1 - \varepsilon$, whenever (2) holds. To prove this, we will take the following path, which is essentially an induction on $r$. For $j \in [p]$, let $b_j = 1$, $dep(\mathbf{C}_j) = \mathbf{C}_j \cup (\mathbf{Pa}(\mathbf{C}_j) \setminus \mathbf{T})$ and $Y_j(\cdot) = 1$ (a constant function). Set $\mathbf{b} = (b_1, \ldots, b_p)$, $\mathbf{dep} := (dep(1), \ldots, dep(r), dep(\mathbf{C}_1), \ldots, dep(\mathbf{C}_p))$, and $\mathbf{Y} = (Y_1, \ldots, Y_p)$.

In Definition 13, we define an optimization program, $P_{r,p}(\Sigma, \gamma, \mathcal{C}, \mathbf{b}, \mathbf{dep}, \mathbf{Y})$ whose objective value is equal to $BC(P_{\mathcal{M}}[\mathbf{W} \mid do(\mathbf{t})], P_{\mathcal{N}}[\mathbf{W} \mid do(\mathbf{t})])$. In Appendix G, we provide the steps to prove a lower bound on the objective of the program, thereby proving a lower bound on $BC(P_{\mathcal{M}}[\mathbf{W} \mid do(\mathbf{t})], P_{\mathcal{N}}[\mathbf{W} \mid do(\mathbf{t})])$.

Also, from (2) and (7), for all $j \in [p]$ and for all $\mathbf{w}_{dep(\mathbf{C}_j) \setminus \mathbf{C}_j}$,

$$\sum_{w_{n_{j,1}}} X_{n_{j,1}}(\mathbf{w}_{dep(n_{j,1})}) \sum_{w_{n_{j,2}}} X_{n_{j,2}}(\mathbf{w}_{dep(n_{j,2})}) \ldots \sum_{w_{n_{j,s_j}}} X_{n_{j,s_j}}(\mathbf{w}_{dep(n_{j,s_j})}) \geq 1 - \gamma$$

satisfying (11). Note that $P_{r,p}(\Sigma, \gamma, \mathcal{C}, \mathbf{b}, \mathbf{dep}, \mathbf{Y})$ is a program such that $\max_j |dep(C_j)| \leq \ell(d+1)$. By Lemma 16,

$$BC\left(P_{\mathcal{M}}[\mathbf{W} \mid do(\mathbf{t})], P_{\mathcal{N}}[\mathbf{W} \mid do(\mathbf{t})]\right) \geq \mathsf{Opt}(P_{r,p}) \geq (1 - |\Sigma|^{\ell(d+1)}\gamma)^p.$$

Using this in (7), we get

$$H^2\left(P_{\mathcal{M}}[\mathbf{W} \mid do(\mathbf{t})], P_{\mathcal{N}}[\mathbf{W} \mid do(\mathbf{t})]\right) \leq 1 - (1 - |\Sigma|^{\ell(d+1)}\gamma)^p \leq p\gamma|\Sigma|^{\ell(d+1)} \leq \epsilon. \qquad \square$$

## F   Proofs for Lower Bound on Interventional Complexity

### F.1   Proof of Lemma 4

**Lemma 4.** *There exists a $G$, and a constant $c$ such that for any set of interventions $\mathbf{I}$ with $|\mathbf{I}| < c \cdot K^{\ell d - 2} \log n$, there is a $\mathbf{C} \subseteq \mathbf{B}$, which is a c-component of $G$, and an assignment $\mathbf{pa}(\mathbf{C})$ such that no intervention in $\mathbf{I}$*

- *assigns $\mathbf{pa}(\mathbf{C})$ to $\mathbf{Pa}(\mathbf{C})$, and*
- *observes all variables in $\mathbf{C}$.*

*Proof.* We show existence of such a $G$ using a probabilistic argument. We consider $\mathbf{A} = \mathbf{A}_r \cup \mathbf{A}_f$, where $\mathbf{A}_r := \{A_1, \ldots, A_n\}$, and $\mathbf{A}_f := \{A_{n+1}, \ldots, A_{n+(\ell d)-2}\}$. We consider $\mathbf{B} := \mathbf{B}_1 \cup \mathbf{B}_2 \cup \ldots \cup \mathbf{B}_{n/\ell}$, where for each $i \in [n/\ell]$, $\mathbf{B}_i = \{B_{i,1}, B_{i,2}, \ldots, B_{i,\ell}\}$. $\mathbf{V} = \mathbf{A} \cup \mathbf{B}$ will be the set of observable nodes in the graph. Therefore, the number of nodes is $|\mathbf{V}| = 2n + \ell d - 2 = O(n)$.

The set of unobservable nodes are such that the following is satisfied:

- $\mathbf{B}_i$ is a c-component in $G$, for each $\mathbf{B}_i$.

We consider random directed bipartite graphs on $\mathbf{V}$ generated as follows, where all the edges go from $\mathbf{A}$ to $\mathbf{B}$. Each c-component $\mathbf{B}_i$ has exactly $\ell d$ parents, chosen as follows:

- $\mathbf{A}_f \subset \mathbf{Pa}(\mathbf{B}_i)$, namely every vertex of $\mathbf{A}_f$ is the parent of at least one node in $\mathbf{B}_i$.

- The remaining two parents of $\mathbf{B}_i$ are chosen randomly from $\mathbf{A}_r$ with edge density $p := 2/n$.

Let $\mathbf{I}$ be a set of interventions that satisfies the conditions of Lemma 3. Let $\mathbf{I}' \subseteq \mathbf{I}$ be the interventions that intervene *all the nodes* in $\mathbf{A}$. The nodes in $\mathbf{A}_f$ can be intervened in $|\Sigma|^{|\mathbf{A}_f|} = K^{\ell d - 2}$ ways. This induces a partition of $\mathbf{I}'$ into $K^{\ell d - 2}$ parts, where the interventions in each partition intervenes $\mathbf{A}_f$ with the same assignment. Let $\{\mathbf{I}_1, \ldots, \mathbf{I}_{K^{\ell d - 2}}\}$ such that $\mathbf{I}' = \mathbf{I}_1 \cup \ldots \cup \mathbf{I}_{K^{\ell d - 2}}$ be this partition. We will show that for each $j$, $|\mathbf{I}_j| = \Omega(\log n)$, implying that

$$|\mathbf{I}| \geq |\mathbf{I}'| \geq K^{\ell d - 2} \cdot \Omega(\log n) = \Omega(K^{\ell d - 2} \log n).$$

Consider a $\mathbf{I}_j$, with $|\mathbf{I}_j| = t$. Further, for simplicity we assume that $K = 2$ for this part, and that $\Sigma = \{0, 1\}$. Since all the nodes in $\mathbf{A}_r$ are intervened, consider one such node. For any node in $\mathbf{A}_r$ consider the $t$ bit binary string denoting whether it is intervened with 0 or 1 in the $t$ interventions. This divides the set $\mathbf{A}_r$ into $2^t$ cells $\mathbf{Z}_1, \ldots, \mathbf{Z}_{2^t}$, where two nodes are in the same cell if they are intervened with the identical value by each intervention in $\mathbf{I}_j$. The expected number of pairs of vertices in $\mathbf{Z}_h$ that are both parents of some vertex in $\mathbf{B}$ is $O(p|\mathbf{Z}_h|^2)$. Therefore, the expected number of pairs of vertices that are both parents of some vertex in $\mathbf{B}$ and also belong to the same cell is $O\left(\sum_h p|\mathbf{Z}_h|^2\right)$, which is at least $O(pn^2 2^{-t})$ (since $\sum_h \mathbf{Z}_h = n$). Now for any such pair of vertices $A, A' \in \mathbf{A}_r$ that belong to the same cell, there exists no intervention such that $A = 0$ and $A' = 1$, contradicting to our requirement. Therefore, $pn^2 2^{-t} < 1$ which implies $t$ is at least $\Omega(\log n)$. $\qquad \square$

Now we proceed to prove Lemma 3.

## F.2 Proof of Lemma 3

**Lemma 3.** *Suppose an adaptive algorithm uses a sequence of interventions $\mathbf{I}$ to solve $\mathsf{C2ST}(G, \epsilon)$ or $\mathsf{CL}(G, \epsilon)$. Let $\mathbf{C} \subseteq \mathbf{B}$ be a c-component of $G$. Then, for any assignment $\mathbf{pa}(\mathbf{C}) \in \Sigma^{|\mathbf{Pa}(\mathbf{C})|}$, there is an intervention $I \in \mathbf{I}$ such that the following conditions hold:*

**C1.** *$I$ intervenes $\mathbf{Pa}(\mathbf{C})$ with the corresponding assignment of $\mathbf{pa}(\mathbf{C})$,[9]*

**C2.** *$I$ does not intervene on any node in $\mathbf{C}$.*

*Proof.* In our construction we consider models where $\mathbf{A}$ is assigned $\mathbf{0}^{|\mathbf{A}|}$ with probability one in the observable distribution. In other words, each $A_i \in \mathbf{A}$ takes value 0 with probability one. Consider any intervention $I$ that targets a $\mathbf{A}' \subseteq \mathbf{A}$. Consider the intervention $I'$ that intervenes $\mathbf{A}'$ the same way as $I$, but intervenes the nodes in $\mathbf{A} \setminus \mathbf{A}'$ with 0's. Since there are no incoming arrows to $\mathbf{A}$, the distribution of $I'$ will be the same as $I$. Therefore, we assume that each intervention $I$ we make intervenes *all the vertices* in $\mathbf{A}$.

Suppose there is an algorithm that makes a series of interventions $\mathbf{I}$ that do not satisfy the conditions of Lemma 3. In other words, there exists a c-component $\mathbf{C} \subseteq \mathbf{B}$ and an assignment $\mathbf{pa}(\mathbf{C})$, such that no intervention in $\mathbf{I}$ satisfies **C1** and **C2**. Let $\mathbf{C} = \{V_1, V_2, \ldots, V_\ell\}$ and $\mathbf{Pa}(\mathbf{C}) = \{W_1, W_2, \ldots, W_s\}$.

Let $G'$ be a subgraph of $G$ on the vertices $\mathbf{C} \cup \mathbf{Pa}(\mathbf{C})$ whose edge set satisfies the following:

- $\mathbf{C}$ contains exactly $\ell - 1$ bidirected edges that form a tree.

- each of the parent vertices $W_i$ has exactly one child node in $\mathbf{C}$.

In our construction, we consider models where the distribution on the rest of the vertices of $G$ (*i.e.,* $\mathbf{V} \setminus (\mathbf{C} \cup \mathbf{Pa}(\mathbf{C}))$) will be independent of the distribution on $\mathbf{C} \cup \mathbf{Pa}(\mathbf{C})$. Therefore, we can restrict our focus on $G'$. We will show the existence of two models $\mathcal{M}$ and $\mathcal{N}$ on $G'$ such that:

**S.1** Let $\mathbf{T} \subseteq (\mathbf{C} \cup \mathbf{Pa}(\mathbf{C}))$, $\mathbf{t} \in \Sigma^{|\mathbf{T}|}$. Let $\{\mathbf{C}_1, \ldots, \mathbf{C}_q\}$ be the c-components of the induced graph $G'[\mathbf{C} \setminus \mathbf{T}]$. Suppose under the intervention $do(\mathbf{t})$, the conditions **C1**, or **C2** is not satisfied, then, the distributions over $\mathbf{C} \setminus \mathbf{T}$ in $\mathcal{M}$ and $\mathcal{N}$ are identical under $do(\mathbf{t})$, namely,

$$P_{\mathcal{M}}[\mathbf{C} \setminus \mathbf{T} \mid do(\mathbf{t})] = \prod_i P_{\mathcal{M}}[\mathbf{C}_i \mid do(\mathbf{t})] = \prod_i P_{\mathcal{N}}[\mathbf{C}_i \mid do(\mathbf{t})] = P_{\mathcal{N}}[\mathbf{C} \setminus \mathbf{T} \mid do(\mathbf{t})]$$

where for each $i$, $P_{\mathcal{M}}[\mathbf{C}_i \mid do(\mathbf{t})] = P_{\mathcal{N}}[\mathbf{C}_i \mid do(\mathbf{t})]$ and is a uniform distribution over $\{0,1\}^{|\mathbf{C}_i|}$,

**S.2** $\delta_{TV}(P_{\mathcal{M}}[\mathbf{C} \mid do(\mathbf{pa}(\mathbf{C}))], P_{\mathcal{N}}[\mathbf{C} \mid do(\mathbf{pa}(\mathbf{C}))]) = 1$.[10]

Recall that the sequence of interventions performed by an (adaptive) algorithm is denoted by $\mathbf{I}$. The assignment $\mathbf{pa}(\mathbf{C})$ gets fixed only after the algorithm fixes *all* the interventions in $\mathbf{I}$. However, we know that any intervention in $\mathbf{I}$ belongs to the category S.1. And for each such intervention in $\mathbf{I}$, the corresponding distributions on models $\mathcal{M}$ and $\mathcal{N}$ are equal, and is defined by a set of uniform distributions over the c-components. Therefore, we can construct an adversary that, for each intervention in $\mathbf{I}$ performed by the algorithm (sequentially), outputs a distribution[11] based on S.1. When the algorithm terminates, the assignment $\mathbf{pa}(\mathbf{C})$ gets fixed, and we can show the existence of two models $\mathcal{M}$ and $\mathcal{N}$ such that

- the models agree on all the interventional distributions in $\mathbf{I}$, and all such distributions also match the corresponding distributions that were revealed by the adversary.

- $\delta_{TV}(P_{\mathcal{M}}[\mathbf{C} \mid do(\mathbf{pa}(\mathbf{C}))], P_{\mathcal{N}}[\mathbf{C} \mid do(\mathbf{pa}(\mathbf{C}))]) = 1$.

Moreover, we can construct such an adversary that outputs distributions in the same way, for all the c-components $\mathbf{C} \subseteq \mathbf{B}$ of $G$. Thus, an explicit construction of two models $\mathcal{M}$ and $\mathcal{N}$ on $G'$ that generates distributions according to S.1 and S.2 would conclude our proof. The remainder of the proof is dedicated towards this goal.

Let $\mathbf{U}$ be the set of all unobservable variables in $G'$. Let $\mathbf{U}^{V_i} \subseteq \mathbf{U}$ represent the bidirected edges incident to $V_i$ in $G'$. Also, for each variable $V_i$ we have an additional boolean random variable $R_i$ that provides randomness to $V_i$. All the randomness in the models $\mathcal{M}$ and $\mathcal{N}$ we construct are in the hidden variables $U_i$'s and the $R_i$'s. In other words, the observable variables are a deterministic function of these. The models $\mathcal{M}$ and $\mathcal{N}$ are defined as follows:

1. (a) For each bidirected edge $U_i \in \mathbf{U}$, $U_i$ is a Bern(0.5) random variable in both $\mathcal{M}$, and $\mathcal{N}$.

   (b) In each model, $R_i$'s are also independent Bern(0.5) random variables.

2. For each $i \in [s]$, $W_i = 0$ with probability one in both $\mathcal{M}$, and $\mathcal{N}$.

3. For each $V_i \in \mathbf{C}$, with probability one:

   (a) when $\mathbf{Pa}(V_i)$ is not consistent with $\mathbf{pa}(V_i)$, then $V_i = \mathsf{XOR}(\mathbf{U}^{V_i}, R_i)$ in both both $\mathcal{M}$, and $\mathcal{N}$.

   (b) when $\mathbf{Pa}(V_i)$ is consistent with $\mathbf{pa}(V_i)$ and $i \neq 1$, then $V_i = \mathsf{XOR}(\mathbf{U}^{V_i})$ in both $\mathcal{M}$, and $\mathcal{N}$.

   (c) when $\mathbf{Pa}(V_i)$ is consistent with $\mathbf{pa}(V_i)$ and $i = 1$, $V_i$ takes
   - $V_i = \mathsf{XOR}(\mathbf{U}^{V_i})$ in $\mathcal{M}$, and
   - $V_i = \mathsf{XNOR}(\mathbf{U}^{V_i})$ in $\mathcal{N}$.

**Case 1: When $I$ respects S.1.** Consider an intervention $I$, identified by $do(\mathbf{t})$, that respects S.1. That is, either $I$ intervenes some node in $\mathbf{C}$, or $I$ does not intervene $\mathbf{Pa}(\mathbf{C})$ with the assignment $\mathbf{pa}(\mathbf{C})$. Let $\{\mathbf{C}_1, \ldots, \mathbf{C}_q\}$ be the c-components of the graph induced by $\mathbf{C} \setminus \mathbf{T}$. Note that the models $M$, and $N$, differ only on the function $V_1$. Therefore, when $V_1$ is intervened in $I$, it is easy to see that the required distributions are equal, and is a product of uniform distributions over the c-components.[12] Suppose $V_1$ is not intervened in $I$, and without loss of generality let $\mathbf{C}_1$ be the c-component that contains $V_1$. Since the models differ only on $V_1$, it is easy to see that $P_{\mathcal{M}}[\mathbf{C}_i \mid do(\mathbf{t})] = P_{\mathcal{N}}[\mathbf{C}_i \mid do(\mathbf{t})]$ for all $i \neq 1$, and is uniform over $\{0,1\}^{|\mathbf{C}_i|}$. Hence, it is sufficient to prove that $P[\mathbf{C}_1 \mid do(\mathbf{t})]$'s are equal and uniform in both models. Let $\mathbf{S}$ be the set of all $U_i$'s and $R_j$'s of the following type: a) $U_i$'s that have one child in $\mathbf{C}_1$ and another child in $\mathbf{T}$; b) $R_j$'s with respect to $V_j \in \mathbf{C}_1$ such that $\mathbf{pa}(V_j)$[13] is inconsistent with $\mathbf{t}$ (*i.e.*, $V_j \in \mathbf{C}_1$ that computes $\mathsf{XOR}(\mathbf{U}^{V_j}, R_j)$). Now, for any fixed assignment $\mathbf{a}_{\mathbf{C}_1^{-1}}$ to $\mathbf{C}_1 \setminus \{V_1\}$ and $\mathbf{a}_{\mathbf{S}}$ to $\mathbf{S}$, because the bi-directed edges within $\mathbf{C}_1$ form a '*tree*', the value of every unobservable variable within $\mathbf{C}_1$[14] can be computed. Note that $V_1$ computes $\mathsf{XOR}(\mathbf{a}_{\mathbf{C}_1^{-1}}, \mathbf{a}_S)$ in $\mathcal{M}$, and $\mathsf{XNOR}(\mathbf{a}_{\mathbf{C}_1^{-1}}, \mathbf{a}_S)$ in $\mathcal{N}$. However, we know that $\mathbf{S}$ is a non-empty set and the bit parities of $\mathbf{S}$ are uniformly distributed in both the models. This implies $P_{\mathcal{M}}[\mathbf{C}_1 \mid do(\mathbf{t})] = P_{\mathcal{N}}[\mathbf{C}_1 \mid do(\mathbf{t})]$, and is a uniform distribution over $\{0,1\}^{|\mathbf{C}_1|}$.

**Case 2: When $I$ respects S.2.** Consider an intervention $I$ that respects S.2. That is, $\mathbf{Pa}(\mathbf{C})$ is intervened with the assignment $\mathbf{pa}(C)$ in $I$, and no node of $\mathbf{C}$ is intervened in $I$. Consider the set of variables $\mathbf{S}$ as defined before for the S.1 case. Note that $\mathbf{S}$ is empty here. This implies, for any fixed assignment $\mathbf{a}_{\mathbf{C}^{-1}}$ to $\mathbf{C} \setminus \{V_1\}$, $V_1$ computes $\mathsf{XOR}(\mathbf{a}_{\mathbf{C}^{-1}})$ in $\mathcal{M}$, and $V_1$ computes $\mathsf{XNOR}(\mathbf{a}_{\mathbf{C}^{-1}})$ in $\mathcal{N}$. This implies, the supports of $P_{\mathcal{M}}[\mathbf{C} \mid do(\mathbf{pa}(\mathbf{C}))]$ and $P_{\mathcal{N}}[\mathbf{C} \mid do(\mathbf{pa}(\mathbf{C}))]$ are disjoint, and therefore the total variation distance is 1.

Hence, irrespective of the number of samples taken from the interventions of $\mathbf{I}$, any adaptive algorithm that solves $\mathsf{C2ST}(G, \epsilon)$ or $\mathsf{CL}(G, \epsilon)$ must consider a sequence of interventions that satisfies the conditions **C1** and **C2**.

$\square$

# G  Program $P_{r,p}$ and Properties

In this section, we gather the technical tools used to prove the subadditivity result, Theorem 3. We formulate our claims at a higher level of abstraction than needed for our purposes, so that the essence of the argument becomes clearer. An illustration of the proof of the subadditivity theorem on a simple causal graph with four vertices can be found in Appendix H.

We begin by defining the optimization problem, and then describe it at a high level.

**Definition 13** (**Program $P_{r,p}(\Sigma, \gamma, \mathcal{C}, \mathbf{b}, \mathbf{dep}, \mathbf{Y})$**)**. *For integers $r, p \geq 0$, suppose the following are given:*

1. *an alphabet set $\Sigma$,*

2. *$\gamma \in (0,1)$,*

3. *a partition[15] $\mathcal{C}$ of $[r]$ into $\mathbf{C}_1, \mathbf{C}_2, \ldots, \mathbf{C}_p$, where for each $j \in [p]$, $s_j = |\mathbf{C}_j|$ and the elements of $\mathbf{C}_j$ are $\{n_{j,1}, \ldots, n_{j,s_j}\}$ in increasing order,*

4. *a vector $\mathbf{b} = (b_1, b_2, \ldots, b_p) \in [0,1]^p$,*

5. *a vector of sets* $\mathbf{dep} = (dep(1), \ldots, dep(r), dep(\mathbf{C}_1), \ldots, dep(\mathbf{C}_p))$ *such that:*

$$[n_{j,i}] \supseteq dep(n_{j,i}) \supseteq \{n_{j,i}\} \cup dep(n_{j,i-1}) \qquad \forall j \in [p], i \in [s_j]$$
$$s_j \neq 0 \implies dep(\mathbf{C}_j) \supseteq dep(n_{j,s_j}) \qquad \forall j \in [p]$$
$$s_j = 0 \implies dep(\mathbf{C}_j) = \emptyset \qquad \forall j \in [p]$$

6. *a set of functions* $\mathbf{Y} = (Y_1, Y_2, \ldots, Y_p)$, *where* $Y_j \colon \Sigma^{|dep(\mathbf{C}_j)|} \to [0,1]$.

*The program* $P_{r,p}(\Sigma, \gamma, \mathcal{C}, \mathbf{b}, \mathbf{dep}, \mathbf{Y})$ *is the following optimization problem over* $\mathbf{X} = (X_1, \ldots, X_r)$ *where* $X_i \colon \Sigma^{|dep(i)|} \to [0,1]$:

$$\min_{\mathbf{X}} f_{r,p}(\mathbf{X}) \overset{\text{def}}{=} \sum_{a_1 \in \Sigma} X_1(\mathbf{a}_{dep(1)}) \sum_{a_2 \in \Sigma} X_2(\mathbf{a}_{dep(2)}) \quad \cdots \sum_{a_r \in \Sigma} X_r(\mathbf{a}_{dep(r)}) \cdot \prod_{j=1}^{p} Y_j(\mathbf{a}_{dep(\mathbf{C}_j)})$$

*subject to*

$$\sum_{a_i \in \Sigma} X_i(\mathbf{a}_{dep(i)}) \leq 1 \qquad \forall i \in [r], \forall \mathbf{a}_{dep(i) \setminus \{i\}} \in \Sigma^{|dep(i) \setminus \{i\}|}$$

$$\tag{10}$$

$$\sum_{a_{n_{j,1}} \in \Sigma} X_{n_{j,1}}(\mathbf{a}_{dep(n_{j,1})}) \sum_{a_{n_{j,2}} \in \Sigma} X_{n_{j,2}}(\mathbf{a}_{dep(n_{j,2})}) \cdots \sum_{a_{n_{j,s_j}} \in \Sigma} X_{n_{j,s_j}}(\mathbf{a}_{dep(n_{j,s_j})}) \cdot Y_j(\mathbf{a}_{dep(\mathbf{C}_j)})$$

$$\geq 1 - b_j \gamma \qquad \forall j \in [p], \forall \mathbf{a}_{dep(\mathbf{C}_j) \setminus \mathbf{C}_j}$$

$$\tag{11}$$

We will first describe the correspondence between the interventional formation of (8) and the above program. Consider the following program where: for $j \in [p]$, $b_j = 1$, $dep(\mathbf{C}_j) = \mathbf{C}_j \cup (\mathbf{Pa}(\mathbf{C}_j) \setminus \mathbf{T})$, $Y_j(\cdot) = 1$ (a constant function), $\mathbf{b} = (b_1, \ldots, b_p)$, $\mathbf{dep} := (dep(1), \ldots, dep(r), dep(\mathbf{C}_1), \ldots, dep(\mathbf{C}_p))$, and $\mathbf{Y} = (Y_1, \ldots, Y_p)$. Substitution of the variables of the program $X_1, X_2, \ldots, X_n$ by the corresponding Bhattacharya coefficients (as defined in (9)) will satisfy all the constraints (10) and (11) of the program: (10) captures the fact that the Bhattacharyya coefficient is at most one; (11) captures the closeness constraint in Theorem 3, *i.e.,* (2). Also, the objective function $f_{r,p}$ captures the required Bhattacharya coefficient of (8). Hence, proving a lower bound on the objective value of this program will suffice to prove Theorem 3. The remainder of this section is dedicated towards this goal.

The next three lemmas (Lemmas 12, 13 and 14) all have the following flavor:

- They take as input an optimization problem (program $P_{r,p}$), and output a new program $P_{r',p}^{\text{new}}$.
- The optimal value of the program only goes down.
- The new program is *simpler* to analyze.[16]

We pass the original program $P_{r,p}$ through the first lemma, and pass its output through the second. The second lemma is applied multiple times until the output program satisfies a particular property. The obtained program is then passed through the third lemma to obtain a new program $P_{r-1,p}$ (with a reduced value of $r$), and the steps repeat[17]. The above procedure reduces to a program with $r = 0$, namely to a program of the form $P_{0,p}$. We can lower bound the objective of this program by simply using (11). Combining these will yield a lower bound on the optimum of the original program $P_{r,p}$, thus proving Theorem 3.

The first lemma takes a program as input and outputs a new program with a smaller optimal value that satisfies $dep(r) = dep(\mathbf{C}_f)$ (where $r \in \mathbf{C}_f$). Let $\mathsf{Opt}(P_{r,p})$ denote the optimal value of the program $P_{r,p}$.

**Lemma 12** (Dependent Set Reduction). *Suppose* $r \in \mathbf{C}_f$. *Let* $P_{r,p}^{\text{new}}$ *be the program obtained from* $P_{r,p}$ *by replacing* $dep(r)$ *by* $dep(\mathbf{C}_f)$, *then*

$$\mathsf{Opt}(P_{r,p}) \geq \mathsf{Opt}(P_{r,p}^{\text{new}}).$$

*Proof.* Our goal is to reduce the given program $P_{r,p}$ to a different program $P_{r,p}^{\text{new}}$ such that $\mathsf{Opt}(P_{r,p}) \geq \mathsf{Opt}(P_{r,p}^{\text{new}})$, where $P_{r,p}^{\text{new}}$ is defined from $P_{r,p}$ by defining $dep(r)$ to be $dep(\mathbf{C}_f)$.

Let $\mathbf{X}^{\text{old}} = \{X_1^{\text{old}}, \ldots X_r^{\text{old}}\}$ be an optimal solution of $P_{r,p}$. Now we construct a *feasible* solution $\mathbf{X}^{\text{new}} = \{X_1^{\text{new}}, \ldots, X_r^{\text{new}}\}$ for the program $P_{r,p}^{\text{new}}$, such that $f_{r,p}^{\text{new}}(\mathbf{X}^{\text{new}}) = f_{r,p}(\mathbf{X}^{\text{old}}) = \mathsf{Opt}(P_{r,p})$. For all $i \neq r$, we define $X_i^{\text{new}} = X_i^{\text{old}}$. For $i = r$, we define $X_r^{\text{new}}(\mathbf{a}_{dep^{\text{new}}(r)}) = X_r^{\text{old}}(x_{dep(r)})$. In other words, $X_r^{\text{new}}$ ignores the new variables added to $dep^{\text{new}}(r)$. Therefore,

$$f_{r,p}^{\text{new}}(\mathbf{X}^{\text{new}}) = \sum_{a_1 \in \Sigma} X_1^{\text{new}}(\mathbf{a}_{dep(1)}) \cdots \sum_{a_r \in \Sigma} X_r^{\text{new}}(\mathbf{a}_{dep^{\text{new}}(r)}) \cdot \prod_{j=1}^{p} Y_j(\mathbf{a}_{dep(\mathbf{C}_j)})$$

$$= \sum_{a_1 \in \Sigma} X_1^{\text{old}}(\mathbf{a}_{dep(1)}) \cdots \sum_{a_r \in \Sigma} X_r^{\text{old}}(\mathbf{a}_{dep(r)}) \cdot \prod_{j=1}^{p} Y_j(\mathbf{a}_{dep(\mathbf{C}_j)}) \text{ (by definition of } \mathbf{X}^{\text{new}})$$

$$= f_{r,p}(X^{\text{old}}) \qquad \text{(by the definition of } f_{r,p}).$$

For the program $P_{r,p}^{\text{new}}$, when $i \neq r$, $\mathbf{X}^{\text{new}}$ satisfies the constraints in (10) (since the functions $X_i^{\text{old}}$ and $X_i^{\text{new}}$ are the same). Similarly, for $j \neq f$, constraints in (11) of the program $P_{r,p}^{\text{new}}$ are valid. When $i = r$ in (10), for each $\mathbf{a}_{dep^{\text{new}}(r)\setminus\{r\}}$, we get

$$\sum_{a_r} X_r^{\text{new}}(\mathbf{a}_{dep^{\text{new}}(r)}) = \sum_{\mathbf{a}_r} X_r^{\text{old}}(\mathbf{a}_{dep(r)}) \leq 1.$$

When $j = f$ in (11), for all $\mathbf{a}_{dep(\mathbf{C}_j)\setminus\mathbf{C}_j}$, since $n_{f,s_f} = r$ we get,

$$\sum_{a_{n_{f,1}} \in \Sigma} X_{n_{f,1}}^{\text{new}}(\mathbf{a}_{dep(n_{f,1})}) \cdots \sum_{a_r \in \Sigma} X_r^{\text{new}}(\mathbf{a}_{dep^{\text{new}}(r)}) \cdot Y_f(\mathbf{a}_{dep(\mathbf{C}_f)})$$

$$= \sum_{a_{n_{f,1}} \in \Sigma} X_{n_{f,1}}^{\text{old}}(\mathbf{a}_{dep(n_{f,1})}) \cdots \sum_{a_r \in \Sigma} X_r^{\text{old}}(\mathbf{a}_{dep(r)}) \cdot Y_f(\mathbf{a}_{dep(\mathbf{C}_f)}) \quad \text{(from definition of } \mathbf{X}^{\text{new}})$$

$$\geq 1 - b_f\gamma \qquad \text{(using (11))}.$$

This implies $X^{\text{new}}$ is a feasible solution for $P_{r,p}^{\text{new}}$ and hence $\mathsf{Opt}(P_{r,p}) \geq \mathsf{Opt}(P_{r,p}^{\text{new}})$. $\qquad\square$

The next lemma takes a program $P_{r,p}$ as input and outputs a new program (with a smaller optimal value) that satisfies $r \notin dep(\mathbf{C}_h)$ (for some given $\mathbf{C}_h$ such that $r \notin \mathbf{C}_h$).

**Lemma 13** (Y-R Reduction). *Let $P_{r,p}(\Sigma, \gamma, \mathcal{C}, \mathbf{b}, \mathbf{dep}, \mathbf{Y})$ be a given program, and there exists $h \in [p]$ such that $r \notin \mathbf{C}_h$ and $r \in dep(\mathbf{C}_h)$. Then, there exists a program $P_{r,p}^{new}(\Sigma, \gamma, \mathcal{C}, \mathbf{b}^{new}, \mathbf{dep}^{new}, \mathbf{Y}^{new})$ such that*

$$\mathsf{Opt}(P_{r,p}) \geq \mathsf{Opt}(P_{r,p}^{new}),$$

*where*

1.  $b_h^{new} = |\Sigma| \cdot b_h$

2.  $dep^{new}(\mathbf{C}_h) = dep(\mathbf{C}_h) \setminus \{r\}$

3.  $b_j^{new} = b_j \qquad\qquad\qquad \forall j \in [p] \setminus \{h\}$

4.  $dep^{new}(C_j) = dep(C_j) \qquad \forall j \in [p] \setminus \{h\}$

5.  $dep^{new}(i) = dep(i) \qquad\quad \forall i \in [r]$

6.  $Y_j^{new}(\mathbf{a}_{dep(\mathbf{C}_j)}) = Y_j(\mathbf{a}_{dep(\mathbf{C}_j)}) \quad \forall j \in [p] \setminus \{h\}, \forall \mathbf{a}_{dep(\mathbf{C}_j)}.$

*Proof.* Let $\mathbf{X}'$ be an optimal solution of $P_{r,p}$. Note that, since $dep(\mathbf{C}_h^{\text{new}}) = dep(\mathbf{C}_h) \setminus \{r\}$, our goal is to find a function $Y_h^{\text{new}} : \Sigma^{|dep(\mathbf{C}_h^{\text{new}})|} \to [0,1]$, whose domain size is smaller than the domain size of $Y_h$ (as $Y_h^{\text{new}}$ is independent of the value of $a_r$), that satisfies the required constraints.

For a given set of functions $\mathbf{X}$, a subset $\mathbf{S} \subseteq [r]$, and for a given assignment $a_\mathbf{S}$ to $\mathbf{S}$, let $f_{r,p}(\mathbf{X})|_{a_\mathbf{S}}$ represent the sum of all terms in $f_{r,p}(\mathbf{X})$ that are consistent with the assignment $a_\mathbf{S}$. Note that

$$f_{r,p}(\mathbf{X}') = \sum_{\mathbf{a}_{dep(\mathbf{C}_h)}} f_{r,p}(\mathbf{X}')|_{\mathbf{a}_{dep(\mathbf{C}_h)}}. \tag{12}$$

For each $\mathbf{a}_{dep(\mathbf{C}_h)\setminus\{r\}}$, let

1. $z_h(\mathbf{a}_{dep(\mathbf{C}_h)\setminus\{r\}}) = \arg\min_{a_r} Y_h(\mathbf{a}_{dep(\mathbf{C}_h)\setminus\{r\}}, a_r)$,

2. $Y_h^{\text{new}}(\mathbf{a}_{dep^{\text{new}}(\mathbf{C}_h)}) = Y_h^{\text{new}}(\mathbf{a}_{dep(\mathbf{C}_h)\setminus\{r\}}) = Y_h(\mathbf{a}_{dep(\mathbf{C}_h)\setminus\{r\}}, z_h(\mathbf{a}_{dep(\mathbf{C}_h)\setminus\{r\}}))$.

Based on the above definition of $Y_h^{\text{new}}$, we know that $f_r^{\text{new}}(\mathbf{X}') \leq f_{r,p}(\mathbf{X}')$. In the remainder of the proof, we show that $\mathbf{X}'$ is also a feasible solution for $P_{r,p}^{\text{new}}$. The first set of constraints of $P_{r,p}^{\text{new}}$ are valid (as we have not modified $\mathbf{X}$). Similarly, the second set of constraints is valid for all $j \neq h$ (as we have not changed any parameters). Now we prove the constraints in (11), for $j = h$. For all assignments $a_{dep^{\text{new}}(\mathbf{C}_h)\setminus\mathbf{C}_h}$,

$$\sum_{a_{n_{h,1}}} X'_{n_{h,1}}(\mathbf{a}_{dep(n_{h,1})}) \cdots \sum_{a_{n_{h,s_h}}} X'_{n_{h,s_h}}(\mathbf{a}_{dep(n_{h,s_h})}) \cdot Y_h^{\text{new}}(\mathbf{a}_{dep^{\text{new}}(C_h)})$$

$$= \sum_{a_{n_{h,1}}} X'_{n_{h,1}}(\mathbf{a}_{dep(n_{h,1})}) \cdots \sum_{a_{h_{s_j}}} X'_{n_{h,s_h}}(\mathbf{a}_{dep(n_{h,s_h})}) \cdot Y_h(\mathbf{a}_{dep(\mathbf{C}_h)\setminus\{r\}}, a_r = z_h(\mathbf{a}_{dep(\mathbf{C}_h)\setminus\{r\}}))$$

$$\text{(by definition of } Y_h^{\text{new}})$$

$$= \left[ \sum_{a_{n_{h,1}}} X'_{n_{h,1}}(\mathbf{a}_{dep(n_{h,1})}) \cdots \sum_{a_{h_{s_j}}} X'_{n_{h,s_h}}(\mathbf{a}_{dep(n_{h,s_h})}) \cdot \sum_{a_r} Y_h(\mathbf{a}_{dep(\mathbf{C}_h\setminus\{r\})}, a_r) \right]$$

$$- \left[ \sum_{a_{n_{h,1}}} X'_{n_{h,1}}(\mathbf{a}_{dep(n_{h,1})}) \cdots \sum_{a_{h_{s_j}}} X'_{n_{h,s_h}}(\mathbf{a}_{dep(n_{h,s_h})}) \cdot \sum_{\substack{a_r : a_r \neq \\ z_h(\mathbf{a}_{dep(\mathbf{C}_h)\setminus\{r\}})}} Y_h(\mathbf{a}_{dep(\mathbf{C}_h)\setminus\{r\}}, a_r) \right]$$

$$\geq \left[ \sum_{a_r} \sum_{a_{n_{h,1}}} X'_{n_{h,1}}(\mathbf{a}_{dep(n_{h,1})}) \cdots \sum_{a_{n_{h,s_h}}} X'_{n_{h,s_h}}(\mathbf{a}_{dep(n_{h,s_h})}) \cdot Y_h(\mathbf{a}_{dep(\mathbf{C}_h)\setminus\{r\}}, a_r) \right]$$

$$- \left[ \sum_{a_{n_{h,1}}} X'_{n_{h,1}}(\mathbf{a}_{dep(n_{h,1})}) \cdots \sum_{a_{h_{s_j}}} X'_{n_{h,s_h}}(\mathbf{a}_{dep(n_{h,s_h})}) \cdot \sum_{\substack{a_r : a_r \neq \\ z(\mathbf{a}_{dep(\mathbf{C}_h)\setminus\{r\}})}} 1 \right] \quad (\because Y_h(.) \leq 1)$$

$$\geq \left[ \sum_{a_r \in \Sigma} (1 - b_h \gamma) \right] - \left[ (|\Sigma| - 1) \cdot \sum_{a_{n_{h,1}}} X'_{n_{h,1}}(\mathbf{a}_{dep(n_{h,1})}) \cdots \sum_{a_{h_{s_j}}} X'_{n_{h,s_h}}(\mathbf{a}_{dep(n_{h,s_h})}) \right]$$

$$\text{(by constraint (11) of } P_{r,p})$$

$$\geq |\Sigma|(1 - b_h \gamma) - (|\Sigma| - 1)1 \quad \text{(by constraint (10) of } P_{r,p})$$
$$= 1 - |\Sigma| b_h \gamma$$
$$= 1 - b_h^{\text{new}} \gamma.$$

$$\square$$

After multiple passes through the above lemma, we get a program $P_{r,p}$ that satisfies $r \notin dep(\mathbf{C}_j)$, for all $\mathbf{C}_j$ such that $r \notin \mathbf{C_j}$. The next lemma takes in such a program, and outputs a program with a reduced value of $r$.

**Lemma 14** (R-Elimination). *Let $P_{r,p}(\Sigma, \gamma, \mathcal{C}, \mathbf{b}, \mathbf{dep}, \mathbf{Y})$ be a given program such that the element $r \in \mathbf{C}_f$. Suppose $dep(r) = dep(\mathbf{C}_f)$, and for all $j \in [p] \setminus \{f\}$, $r \notin dep(\mathbf{C}_j)$. Then there exists a program $P_{r-1,p}^{new}(\Sigma, \gamma, \mathcal{C}^{new}, \mathbf{b}, \mathbf{dep}^{new}, \mathbf{Y}^{new})$ such that*

$$\mathsf{Opt}(P_{r,p}) \geq \mathsf{Opt}(P_{r-1,p}^{new})$$

*where $\mathbf{Y}^{new}$ differs from $\mathbf{Y}$ only on the function $Y_f$, $\mathcal{C}^{new}$ differs from $\mathcal{C}$ only on the partition $\mathbf{C}_f$ where $\mathbf{C}_f^{new} = \mathbf{C}_f \setminus \{r\}$, and $\mathbf{dep}^{new} = (dep(1), dep(2), \ldots, dep(r-1), dep(\mathbf{C}_1), \ldots, dep(\mathbf{C}_{f-1}), dep^{new}(\mathbf{C}_f^{new}), dep(\mathbf{C}_{f+1}), \ldots, dep(\mathbf{C}_p))$ where $dep^{new}(\mathbf{C}_f^{new}) = dep(r) \setminus \{r\}$.*

*Proof.* Let $\mathbf{X}^{old}$ be an optimal solution of $P_{r,p}$. For a given set of functions $\mathbf{X}$, a subset $\mathbf{S} \subseteq [r]$, and for a given assignment $a_{\mathbf{S}}$ to $\mathbf{S}$, let $f_{r,p}(\mathbf{X})|_{a_{\mathbf{S}}}$ represent the sum of all terms in $f_{r,p}(\mathbf{X})$ that are consistent with the assignment $a_{\mathbf{S}}$. Then, for all assignments $\mathbf{a}_{dep(r)\setminus\{r\}}$

$$f_{r,p}(\mathbf{X}^{old})|_{\mathbf{a}_{dep(r)\setminus\{r\}}} = L_{\mathbf{a}_{dep(r)\setminus\{r\}}} \cdot \sum_{a_r} X_r^{old}(\mathbf{a}_{dep(r)}) \cdot Y_f(\mathbf{a}_{dep(r)}) \tag{13}$$

We define $Y_f^{new}(\mathbf{a}_{dep^{new}(\mathbf{C}_f^{new})}) = Y_f^{new}(\mathbf{a}_{dep(r)\setminus\{r\}}) = \sum_{a_r} X_r^{old}(\mathbf{a}_{dep(r)}) \cdot Y_f(\mathbf{a}_{dep(r)})$. Observe that $Y_f^{new} : \Sigma^{|dep(r)\setminus\{r\}|} \to [0,1]$ because of constraint (10) and since $Y_f$ itself falls in the range $[0,1]$. Now, the new program $P_{r-1,p}^{new}$ is completely specified.

Observe that:

$$f_{r,p}(\mathbf{X}^{old})|_{\mathbf{a}_{dep(r)\setminus\{r\}}} = L_{\mathbf{a}_{dep(r)\setminus\{r\}}} \cdot Y_f^{new}(\mathbf{a}_{dep(r)\setminus\{r\}}) = f_{r-1,p}^{new}(\mathbf{X}_{r-1}^{old})|_{\mathbf{a}_{dep(r)\setminus\{r\}}}$$

where $\mathbf{X}_{r-1}^{old} = \{X_1^{old}, \ldots, X_{r-1}^{old}\}$. This implies

$$f_{r,p}(\mathbf{X}^{old}) = \sum_{\mathbf{a}_{dep(r)\setminus\{r\}}} f_{r,p}(\mathbf{X}^{old})|_{\mathbf{a}_{dep(r)\setminus\{r\}}} = \sum_{\mathbf{a}_{dep(r)\setminus\{r\}}} f_{r-1,p}^{new}(\mathbf{X}_{r-1}^{old})|_{\mathbf{a}_{dep(r)\setminus\{r\}}} = f_{r-1,p}^{new}(\mathbf{X}_{r-1}^{old}).$$

We now show that the functions $\mathbf{X}_{r-1}^{old}$ form a feasible solution for $P_{r-1,p}^{new}$. The first set of constraints (10) holds for $P_{r-1,p}^{new}$ because $\mathbf{X}^{old}$ is feasible for $P_{r,p}$. Also for all $j \neq f$, the second set of constraints (11) holds for the same reason. For $j = f$:

$$\sum_{a_{n_{f,1}}} X_{n_{f,1}}^{old}(\mathbf{a}_{dep(n_{f,1})}) \cdots \sum_{a_{n_{f,s_f-1}}} X_{n_{f,s_f-1}}^{old}(\mathbf{a}_{dep(n_{f,s_f-1})}) \cdot Y_f^{new}(\mathbf{a}_{dep(C_f^{new})})$$

$$= \sum_{a_{n_{f,1}}} X_{n_{f,1}}^{old}(\mathbf{a}_{dep(n_{f,1})}) \cdots \sum_{a_{n_{f,s_f-1}}} X_{n_{f,s_f-1}}^{old}(\mathbf{a}_{dep(n_{f,s_f-1})}) \sum_{a_r} X_r^{old}(\mathbf{a}_{dep(r)}) \cdot Y_f(\mathbf{a}_{dep(\mathbf{C}_f)})$$

$$\text{(by definition)}$$

$$\geq 1 - b_f \gamma$$

This completes the proof that $\mathsf{Opt}(P_{r,p}) \geq \mathsf{Opt}(P_{r-1,p}^{new})$. $\qquad\square$

**Lemma 15.** *For any integers $r, p \geq 1$ and given a program $P_{r,p}(\Sigma, \gamma, \mathcal{C}, \mathbf{b}, \mathbf{dep}, \mathbf{Y})$, there exists a program $P_{r-1,p}^{new}(\Sigma, \gamma, \mathcal{C}^{new}, \mathbf{b}^{new}, \mathbf{dep}^{new}, \mathbf{Y}^{new})$ such that*

$$\mathsf{Opt}(P_{r,p}) \geq \mathsf{Opt}(P_{r-1,p}^{new})$$

*where*

$$
\begin{aligned}
b_j^{new} &= b_j & &, \forall j \in [p]: r \notin dep(\mathbf{C}_j), \\
b_j^{new} &= |\Sigma| \cdot b_j & &, \forall j \in [p]: r \in dep(\mathbf{C}_j).
\end{aligned}
$$

*Proof.* First we apply Lemma 12 (Dependent Set Reduction). Then, we apply Lemma 13 (Y-R Reduction) repeatedly, until there does not exist any $h \in [p]$ such that $r \in dep(\mathbf{C}_h)$ but $r \notin \mathbf{C}_h$. Note that, in each step of this reduction, the respective $b_h$ increases by a factor of $|\Sigma|$. Finally, applying Lemma 14 (R-Elimination) results in a program $P_{r-1,p}^{new}$ on $r-1$ inputs with the desired property. $\quad\square$

**Lemma 16.** *For a given program $P_{r,p}(\Sigma, \gamma, \mathcal{C}, \mathbf{b} = 1, \mathbf{dep}, \mathbf{Y} = 1)$, suppose we know that $\max_j |dep(\mathbf{C}_j)|$ is at most $L$. Then*

$$\mathsf{Opt}(P_{r,p}) \geq (1 - |\Sigma|^L \gamma)^p.$$

*Proof.* We apply Lemma 15 recursively. Note that in each such reduction from $P_{r,p}$ to $P_{r-1,p}$, the value of $b_j$ increases by a factor of $|\Sigma|$ only when $r \in dep(\mathbf{C}_j)$.

At $r = 0$, we have the program $P_{0,p}(\Sigma, \gamma, \mathcal{C}', \mathbf{b}', \mathbf{dep}', \mathbf{Y}')$. For all $j \in [p]$, we know that $b'_j \leq |\Sigma|^L$ (since $|dep(\mathbf{C}_j)| \leq L$). Therefore,

$$
\begin{aligned}
\mathsf{Opt}(P_{r,p}) &\geq \mathsf{Opt}(P_{0,p}) \\
&= \prod_{j=1}^{p} Y'_j(\emptyset) \\
&\geq \prod_{j=1}^{p} (1 - b'_j \gamma) \quad \text{based on constraint (11) of the program } P_{0,p} \\
&\geq (1 - |\Sigma|^L \gamma)^p.
\end{aligned}
$$

$\square$

# H   Illustration

In this section, we illustrate the proof of the subadditivity theorem and the c-component factorization of [TP02] using models defined on the graph $\tilde{G}$ (shown in Figure 1), where the maximum in-degree $d$ and the maximum c-component size $\ell$ of $\tilde{G}$ are bounded by 1 and 2 respectively.

Figure 1: Causal graph $\tilde{G}$

## H.1   Illustration of c-component factorization

### H.1.1   Rule 3 of do-calculus

First we recall the Rule 3 of do-calculus [Pea09]. Let $\mathcal{M}$ be a SMBN defined on some SMCG G and let $\mathbf{W}, \mathbf{X}, \mathbf{Y}, \mathbf{Z}$ be disjoint sets of observable variables. Then, for any assignment $\mathbf{w}, \mathbf{x}, \mathbf{y}, \mathbf{z}$, the Rule 3 of do-calculus says that

$$P_{\mathcal{M}}[\mathbf{y} \mid \mathbf{w}, do(\mathbf{z}, \mathbf{x})] = P_{\mathcal{M}}[\mathbf{y} \mid \mathbf{w}, \mathbf{z}, do(\mathbf{x})]$$

if $\mathbf{Y}$ is independent of $\mathbf{Z}$ conditioned on $\mathbf{X}, \mathbf{W}$ in the graph $G_{\overline{\mathbf{X}}, \underline{\mathbf{Z}}}$, where $G_{\overline{\mathbf{X}}, \underline{\mathbf{Z}}}$ is the graph obtained from $G$ by removing the incoming edges to $\mathbf{X}$ and outgoing edges from $\mathbf{Z}$. The conditional independence constraints are based on the well-known d-separation criterion [Pea09].

### H.1.2   C-component Factorization

Now we are ready to illustrate the c-component factorization of [TP02]. Let $\mathcal{M}$ be a SMBN defined on $\tilde{G}$. We will prove that for any assignment $v_1, v_2, v_3, v_4$,

$$P_{\mathcal{M}}[v_1, v_2, v_3, v_4] = P_{\mathcal{M}}[v_1, v_3 \mid do(v_2, v_4)] P_{\mathcal{M}}[v_2, v_4 \mid do(v_1, v_3)].$$

*Proof.* By Bayes' rule:

$$P_{\mathcal{M}}[v_1, v_2, v_3, v_4] = P_{\mathcal{M}}[v_1]P_{\mathcal{M}}[v_2 \mid v_1]P_{\mathcal{M}}[v_3 \mid v_1, v_2]P_{\mathcal{M}}[v_4 \mid v_1, v_2, v_3]. \qquad (14)$$

By definition of SMBN, it directly follows that $P_{\mathcal{M}}[v_1] = P_{\mathcal{M}}[v_1 \mid do(v_2)] = P_{\mathcal{M}}[v_1 \mid do(v_2, v_4)]$ and $P_{\mathcal{M}}[v_2 \mid v_1] = P_{\mathcal{M}}[v_2 \mid do(v_1)] = P_{\mathcal{M}}[v_2 \mid do(v_1, v_3)]$.

Also, the following conditional independence constraints can be verified using d-separation: (i) $V_3$ is independent of $V_2$ conditioned on $V_1$ in the graph $\tilde{G}_{\overline{V_2}}$; (ii) $V_4$ is independent of $V_1, V_3$ conditioned on $V_2$ in the graph $\tilde{G}_{\overline{V_1, V_3}}$. Hence, using the Rule 3 of do-calculus we get

$$P_{\mathcal{M}}[v_3 \mid v_1, v_2] = P_{\mathcal{M}}[v_3 \mid v_1, do(v_2)] = P_{\mathcal{M}}[v_3 \mid v_1, do(v_2, v_4)]$$
$$P_{\mathcal{M}}[v_4 \mid v_1, v_2, v_3] = P_{\mathcal{M}}[v_4 \mid v_2, do(v_1, v_3)].$$

Substituting the above equalities in Equation (14),

$$P_{\mathcal{M}}[v_1, v_2, v_3, v_4] = P_{\mathcal{M}}[v_1 \mid do(v_2)]P_{\mathcal{M}}[v_2 \mid do(v_1, v_3)]P_{\mathcal{M}}[v_3 \mid v_1, do(v_2)]P_{\mathcal{M}}[v_4 \mid v_2, do(v_1, v_3)].$$

Applying Bayes' rule to the above expression yields the desired factorization:

$$P_{\mathcal{M}}[v_1, v_2, v_3, v_4] = P_{\mathcal{M}}[v_1, v_3 \mid do(v_2, v_4)]P_{\mathcal{N}}[v_2, v_4 \mid do(v_1, v_3)].$$

$\square$

### H.2 Illustration of subadditivity theorem

Here we illustrate the proof of the subadditivity theorem (Theorem 3) with respect to models $\mathcal{M}$ and $\mathcal{N}$ defined on $\tilde{G}$ for the case $\mathbf{T} = \emptyset$. Other cases (where $\mathbf{T} \neq \emptyset$) can be easily proved using similar arguments. In particular, we prove the following statement:

**Theorem 9.** *Let $\mathcal{M}$ and $\mathcal{N}$ be two* SMBN*s defined on a* known and common SMCG $\tilde{G}$. *Let* $\mathbf{V}$ *partition into* $\mathcal{C} = \{\mathbf{C}_1 = \{V_1, V_3\}, \mathbf{C}_2 = \{V_2, V_4\}\}$, *the c-components with respect to the induced graph $\tilde{G}$. Suppose*

$$H^2(P_{\mathcal{M}}[V_1, V_3 \mid do(v_2)], P_{\mathcal{N}}[V_1, V_3 \mid do(v_2)]) \leq \gamma \qquad \forall v_2 \in \Sigma \qquad (15)$$
$$H^2(P_{\mathcal{M}}[V_2, V_4 \mid do(v_1, v_3)], P_{\mathcal{N}}[V_2, V_4 \mid do(v_1, v_3)]) \leq \gamma \qquad \forall v_2 \in \Sigma^2. \qquad (16)$$

*Then*

$$H^2\left(P_{\mathcal{M}}[\mathbf{V}], P_{\mathcal{N}}[\mathbf{V}]\right) \leq \epsilon \qquad (17)$$

*where $\epsilon = 2|\Sigma|\gamma$.*

*Proof.* We know that $H^2(P_{\mathcal{M}}[\mathbf{V}], P_{\mathcal{N}}[\mathbf{V}]) = 1 - BC(P_{\mathcal{M}}[\mathbf{V}], P_{\mathcal{N}}[\mathbf{V}])$. By c-component factorization ([TP02]), we get

$$BC(P_{\mathcal{M}}[\mathbf{V}], P_{\mathcal{N}}[\mathbf{V}])$$
$$= \sum_{\mathbf{v}} \sqrt{P_{\mathcal{M}}[\mathbf{v}]P_{\mathcal{N}}[\mathbf{v}]}$$
$$= \sum_{\mathbf{v}} \sqrt{\frac{P_{\mathcal{M}}[v_1, v_3 \mid do(v_2)]P_{\mathcal{M}}[v_2, v_4 \mid do(v_1, v_3)]}{P_{\mathcal{N}}[v_1, v_3 \mid do(v_2)]P_{\mathcal{N}}[v_2, v_4 \mid do(v_1, v_3)]}}$$
$$= \sum_{\mathbf{v}} \sqrt{\frac{P_{\mathcal{M}}[v_1 \mid do(v_2)]P_{\mathcal{M}}[v_3 \mid v_1, do(v_2)]P_{\mathcal{M}}[v_2 \mid do(v_1, v_3)]P_{\mathcal{M}}[v_4 \mid v_2, do(v_1, v_3)]}{P_{\mathcal{N}}[v_1 \mid do(v_2)]P_{\mathcal{N}}[v_3 \mid v_1, do(v_2)]P_{\mathcal{N}}[v_2 \mid do(v_1, v_3)]P_{\mathcal{N}}[v_4 \mid v_2, do(v_1, v_3)]}}$$
$$= \sum_{\mathbf{v}} \sqrt{\frac{P_{\mathcal{M}}[v_1]P_{\mathcal{M}}[v_3 \mid v_1, do(v_2)]P_{\mathcal{M}}[v_2 \mid do(v_1)]P_{\mathcal{M}}[v_4 \mid v_2, do(v_1, v_3)]}{P_{\mathcal{N}}[v_1]P_{\mathcal{N}}[v_3 \mid v_1, do(v_2)]P_{\mathcal{N}}[v_2 \mid do(v_1)]P_{\mathcal{N}}[v_4 \mid v_2, do(v_1, v_3)]}} \qquad (18)$$

Let: $dep(1) = \{1\}$, $dep(2) = \{1, 2\}$, $dep(3) = \{1, 2, 3\}$ and $dep(4) = \{1, 2, 3, 4\}$. Also, let: $dep(\mathbf{C}_1) = \{1, 2, 3\}$ and $dep(\mathbf{C}_2) = \{1, 2, 3, 4\}$. And, for $j \in \{1, 2\}$, let $Y_j : \Sigma^2 \to [0, 1]$ be a function.

Consider the following optimization problem[18] $P_{4,2}$ over $\mathbf{X} = \{X_1, X_2, X_3, X_4\}$ where $X_i : \Sigma^{|dep(i)|} \to [0,1]$:

$$\min_{\mathbf{X}} f_{4,2}(\mathbf{X}) \overset{\text{def}}{=}$$

$$\sum_{v_1 \in \Sigma} X_1(v_1) \sum_{v_2 \in \Sigma} X_2(v_1, v_2) \sum_{v_3 \in \Sigma} X_3(v_1, v_2, v_3) \sum_{v_4 \in \Sigma} X_4(v_1, v_2, v_3, v_4) Y_1(v_1, v_2, v_3) Y_2(v_1, v_2, v_3, v_4)$$

subject to

$$\sum_{v_i \in \Sigma} X_i(\mathbf{v}_{dep(i)}) \leq 1 \qquad \forall i \in [4], \forall \mathbf{v}_{dep(i) \setminus \{i\}} \in \Sigma^{|dep(i) \setminus \{i\}|} \tag{19}$$

$$\sum_{v_1 \in \Sigma} X_1(v_1) \sum_{v_3 \in \Sigma} X_3(v_1, v_2, v_3) \cdot Y_1(v_1, v_2, v_3) \geq 1 - b_1 \gamma \qquad \forall v_2 \in \Sigma \tag{20}$$

$$\sum_{v_2 \in \Sigma} X_2(v_1, v_2) \sum_{v_4 \in \Sigma} X_4(v_1, v_2, v_3, v_4) \cdot Y_2(v_1, v_2, v_3, v_4) \geq 1 - b_2 \gamma \qquad \forall v_1 \in \Sigma, \forall v_3 \in \Sigma \tag{21}$$

where $Y_j(\cdot) = 1$ for both $j \in \{1, 2\}$, and $b_1 = b_2 = 1$.

The instantiation of $P_{4,2}$ with $X_i(\mathbf{v}_{dep(i)}) = \sqrt{P_{\mathcal{M}}[v_i| \cdots, do(\cdots)] P_{\mathcal{N}}[v_i| \cdots, do(\cdots)]}$ of Equation (18) for each $i \in [4]$ satisfies all the constraints (19), (20), (21) of $P_{4,2}$. Hence, any lower bound for the optimal value of the above program $P_{4,2}$ (denoted by $\mathsf{Opt}(P_{4,2})$) will also be a lower bound for $BC(P_{\mathcal{M}}[V], P_{\mathcal{N}}[V])$ (Equation (18)). Therefore, in what follows, we will show a lower bound for $\mathsf{Opt}(P_{4,2})$.

**Step 1 (R-Elimination):** Let $\mathbf{X}^{\text{opt}} = \{X_1^{\text{opt}}, X_2^{\text{opt}}, X_3^{\text{opt}}, X_4^{\text{opt}}\}$ be an optimal solution of $P_{4,2}$. We now define a new program $P_{3,2}^{(\text{I})}$ [19] over three variables such that $\mathsf{Opt}(P_{4,2}) \geq \mathsf{Opt}(P_{3,2}^{(\text{I})})$. The program $P_{3,2}^{(\text{I})}$ is an optimization problem over $\mathbf{X} = \{X_1, X_2, X_3\}$ such that:

$$\min_{\mathbf{X}} f_{3,2}^{(\text{I})}(\mathbf{X}) \overset{\text{def}}{=} \sum_{v_1 \in \Sigma} X_1(v_1) \sum_{v_2 \in \Sigma} X_2(v_1, v_2) \sum_{v_3 \in \Sigma} X_3(v_1, v_2, v_3) Y_1^{(\text{I})}(v_1, v_2, v_3) Y_2^{(\text{I})}(v_1, v_2, v_3)$$

subject to

$$\sum_{v_i \in \Sigma} X_i(\mathbf{v}_{dep^{(\text{I})}(i)}) \leq 1 \qquad \forall i \in [4], \forall \mathbf{v}_{dep^{(\text{I})}(i) \setminus \{i\}} \in \Sigma^{|dep^{(\text{I})}(i) \setminus \{i\}|} \tag{22}$$

$$\sum_{v_1 \in \Sigma} X_1(v_1) \sum_{v_3 \in \Sigma} X_3(v_1, v_2, v_3) \cdot Y_1^{(\text{I})}(v_1, v_2, v_3) \geq 1 - b_1^{(\text{I})} \gamma \qquad \forall v_2 \in \Sigma \tag{23}$$

$$\sum_{v_2 \in \Sigma} X_2(v_1, v_2) \cdot Y_2^{(\text{I})}(v_1, v_2, v_3) \geq 1 - b_2^{(\text{I})} \gamma \qquad \forall v_1, v_3 \in \Sigma^2 \tag{24}$$

where all the parameters of $P_{3,2}^{(\text{I})}$ remain the same as $P_{4,2}$ except the following:

- $dep^{(\text{I})}(\mathbf{C}_2)$ is equal to $\{1, 2, 3\}$.
- For each $v_1, v_2, v_3$: $Y_2^{(\text{I})}(v_1, v_2, v_3) = \sum_{v_4 \in \Sigma} X_4(v_1, v_2, v_3, v_4)$.

By definition, it immediately follows that $X_1^{\text{opt}}, X_2^{\text{opt}}, X_3^{\text{opt}}$ is a feasible solution for the new program $P_{3,2}^{(\text{I})}$, and the objective value $f_{3,2}^{(\text{I})} = f_{4,2}$. Hence it is sufficient to show a lower bound for $\mathsf{Opt}(P_{3,2}^{(\text{I})})$ (Since $BC(P_{\mathcal{M}}[V], P_{\mathcal{N}}[V]) \geq \mathsf{Opt}(P_{4,2}) \geq \mathsf{Opt}(P_{3,2}^{(\text{I})})$).

**Step 2 (Y-R-Reduction):** Note that $3 \notin \mathbf{C}_2$, but $3 \in dep^{\text{I}}(\mathbf{C}_2)$. Hence, we reduce the previous program $P_{3,2}^{(\text{I})}$ to the following new program $P_{3,2}^{(\text{II})}$ that satisfies $3 \notin dep^{(\text{II})}(\mathbf{C}_2)$, such that the optimal value of the new program is smaller than the optimal value of $P_{3,2}^{(\text{I})}$:

$$\min_{\mathbf{X}} f_{3,2}^{(\mathrm{II})}(\mathbf{X}) \stackrel{\text{def}}{=} \sum_{v_1 \in \Sigma} X_1(v_1) \sum_{v_2 \in \Sigma} X_2(v_1, v_2) \sum_{v_3 \in \Sigma} X_3(v_1, v_2, v_3) Y_1^{(\mathrm{II})}(v_1, v_2, v_3) Y_2^{(\mathrm{II})}(v_1, v_2)$$

subject to

$$\sum_{v_i \in \Sigma} X_i(\mathbf{v}_{dep^{(\mathrm{II})}(i)}) \le 1 \qquad \forall i \in [3], \forall \mathbf{v}_{dep^{(\mathrm{II})}(i) \setminus \{i\}} \in \Sigma^{|dep^{(\mathrm{II})}(i) \setminus \{i\}|} \tag{25}$$

$$\sum_{v_1 \in \Sigma} X_1(v_1) \sum_{v_3 \in \Sigma} X_3(v_1, v_2, v_3) \cdot Y_1^{(\mathrm{II})}(v_1, v_2, v_3) \ge 1 - b_1^{(\mathrm{II})} \gamma \qquad \forall v_2 \in \Sigma \tag{26}$$

$$\sum_{v_2 \in \Sigma} X_2(v_1, v_2) \cdot Y_2^{(\mathrm{II})}(v_1, v_2) \ge 1 - b_2^{(\mathrm{II})} \gamma \qquad \forall v_1 \in \Sigma \tag{27}$$

where all the parameters of $P_{3,2}^{(\mathrm{II})}$ remain the same as $P_{3,2}^{(\mathrm{I})}$ except the following:

- $b_2^{(\mathrm{II})} = |\Sigma| b_2^{(\mathrm{I})} = |\Sigma|$ (because $b_2^{(\mathrm{I})} = 1$).
- $dep^{(\mathrm{II})}(\mathbf{C}_2) = \{1, 2\}$.
- For each $v_1, v_2$: $Y_2^{(\mathrm{II})}(v_1, v_2) = Y_2^{(\mathrm{I})}(v_1, v_2, \arg\min_{v_3} Y_2^{(\mathrm{I})}(v_1, v_2, v_3))$.

Let $\mathbf{X}^{\mathrm{opt}} = (X_1^{\mathrm{opt}}, X_2^{\mathrm{opt}}, X_3^{\mathrm{opt}})$ be an optimal solution for $P_{3,2}^{(\mathrm{I})}$. We prove that $\mathbf{X}^{\mathrm{opt}}$ is a feasible solution for $P_{3,2}^{(\mathrm{II})}$. Except for (27), all the other constraints of $P_{3,2}^{(\mathrm{II})}$ can be easily verified. Hence, we prove(27): for each $v_1 \in \Sigma$,

$$\sum_{v_2 \in \Sigma} X_2(v_1, v_2) \cdot Y_2^{(\mathrm{II})}(v_1, v_2)$$

$$= \sum_{v_2 \in \Sigma} X_2(v_1, v_2) \cdot Y_2^{(\mathrm{I})}(v_1, v_2, \arg\min_{v_3} Y_2^{(\mathrm{II})}(v_1, v_2, v_3)) \qquad \text{(by definition of } Y_2^{(\mathrm{II})})$$

$$= \left[ \sum_{v_2 \in \Sigma} X_2(v_1, v_2) \sum_{v_3} Y_2^{(\mathrm{I})}(v_1, v_2, v_3) \right] - \left[ \sum_{v_2 \in \Sigma} X_2(v_1, v_2) \cdot Y_2^{(\mathrm{I})}(v_1, v_2, \arg\min_{v_3} Y_2^{(\mathrm{I})}(v_1, v_2, v_3)) \right]$$

$$\ge \left[ \sum_{v_3 \in \Sigma} (1 - b_2^{(\mathrm{I})} \gamma) \right] - \left[ (|\Sigma| - 1) \cdot \sum_{v_2 \in \Sigma} X_2(v_1, v_2) \right] \qquad \text{(by constraint (24) of } P_{3,2}^{(\mathrm{I})})$$

$$\ge |\Sigma|(1 - b_2^{(\mathrm{I})} \gamma) - (|\Sigma| - 1)1 \quad \text{(by constraint (22) of } P_{3,2}^{(\mathrm{I})})$$

$$= 1 - |\Sigma| b_2^{(\mathrm{I})} \gamma = 1 - b_2^{(\mathrm{II})} \gamma.$$

Also note that $f_{3,2}^{(\mathrm{II})}(\mathbf{X}) = f_{3,2}^{(\mathrm{I})}(\mathbf{X})$. Hence $\mathsf{Opt}(P_{3,2}^{(\mathrm{II})}) \le \mathsf{Opt}(P_{3,2}^{(\mathrm{I})})$.

**Step 3 (R-Elimination):** Since $3 \in \mathbf{C}_1$ and $dep^{(\mathrm{II})}(3) = dep^{(\mathrm{II})}(\mathbf{C}_1)$, $3 \notin \mathbf{C}_2$ and $3 \notin dep^{(\mathrm{II})}(\mathbf{C}_2)$, we can reduce $P_{3,2}^{(\mathrm{II})}$ to a much simpler program $P_{2,2}^{(\mathrm{III})}$ which is defined over only two variables. Let $\mathbf{X}^{\mathrm{opt}} = \{X_1^{\mathrm{opt}}, X_2^{\mathrm{opt}}, X_3^{\mathrm{opt}}\}$ be an optimal solution of $P_{3,2}^{(\mathrm{II})}$. The new program $P_{2,2}^{(\mathrm{III})}$ over $\mathbf{X} = \{X_1, X_2\}$ is defined as follows:

$$\min_{\mathbf{X}} f_{2,2}^{(\mathrm{III})}(\mathbf{X}) \stackrel{\text{def}}{=} \sum_{v_1 \in \Sigma} X_1(v_1) \sum_{v_2 \in \Sigma} X_2(v_1, v_2) Y_1^{(\mathrm{III})}(v_1, v_2) Y_2^{(\mathrm{III})}(v_1, v_2)$$

subject to

$$\sum_{v_i \in \Sigma} X_i(\mathbf{v}_{dep^{(\mathrm{III})}(i)}) \le 1 \qquad \forall i \in [2], \forall \mathbf{v}_{dep^{(\mathrm{III})}(i) \setminus \{i\}} \in \Sigma^{|dep^{(\mathrm{III})}(i) \setminus \{i\}|} \tag{28}$$

$$\sum_{v_1 \in \Sigma} X_1(v_1) \cdot Y_1^{(\mathrm{III})}(v_1, v_2) \ge 1 - b_1^{(\mathrm{III})} \gamma \qquad \forall v_2 \in \Sigma \tag{29}$$

$$\sum_{v_2 \in \Sigma} X_2(v_1, v_2) \cdot Y_2^{(\mathrm{III})}(v_1, v_2) \ge 1 - b_2^{(\mathrm{III})} \gamma \qquad \forall v_1 \in \Sigma \tag{30}$$

where all the parameters of $P_{2,2}^{(\text{III})}$ remain the same as $P_{3,2}^{(\text{II})}$ except the following:

- $dep^{(\text{III})}(\mathbf{C}_1) = \{1, 2\}$.

- For each $v_1, v_2$: $Y_1^{(\text{III})}(v_1, v_2) = \sum_{v_3 \in \Sigma} X_3(v_1, v_2, v_3) Y_1^{(\text{II})}(v_1, v_2, v_3)$.

It directly follows from the definition that $X_1^{\text{opt}}, X_2^{\text{opt}}$ is a feasible solution to the new program $P_{2,2}^{(\text{III})}$, and $f_{2,2}^{(\text{III})}(X_1^{\text{opt}}, X_2^{\text{opt}}) = f_{3,2}^{(\text{II})}(X_1^{\text{opt}}, X_2^{\text{opt}}, X_3^{\text{opt}})$. Therefore, $\mathsf{Opt}(P_{2,2}^{(\text{III})}) \leq \mathsf{Opt}(P_{3,2}^{(\text{II})})$, and hence it is sufficient to show a lower bound for $\mathsf{Opt}(P_{2,2}^{(\text{III})})$ to achieve the desired bound.

**Step 4 (Y-R-Reduction):** Since $2 \notin \mathbf{C}_1$ and $2 \in dep^{(\text{III})}(\mathbf{C}_1)$, we reduce the program $P_{2,2}^{(\text{III})}$ to a new program $P_{2,2}^{(\text{IV})}$ that satisfies $2 \in dep^{(\text{IV})}(\mathbf{C}_1)$, such that optimal value of the new program is lesser than the previous program.

$$\min_{\mathbf{X}} f_{2,2}^{(\text{IV})}(\mathbf{X}) \stackrel{\text{def}}{=} \sum_{v_1 \in \Sigma} X_1(v_1) \sum_{v_2 \in \Sigma} X_2(v_1, v_2) \cdot \prod_{j=1}^{2} Y_j^{(\text{IV})}(\mathbf{v}_{dep^{(\text{IV})}(\mathbf{C}_j)})$$

subject to

$$\sum_{v_i \in \Sigma} X_i(\mathbf{v}_{dep^{(\text{IV})}(i)}) \leq 1 \qquad \forall i \in [2], \forall \mathbf{v}_{dep^{(\text{IV})}(i) \setminus \{i\}} \in \Sigma^{|dep^{(\text{IV})}(i) \setminus \{i\}|} \tag{31}$$

$$\sum_{v_1 \in \Sigma} X_1(v_1) \cdot Y_1^{(\text{IV})}(v_1) \geq 1 - b_1^{(\text{IV})} \gamma \tag{32}$$

$$\sum_{v_2 \in \Sigma} X_2(v_1, v_2) \cdot Y_2^{(\text{IV})}(v_1, v_2) \geq 1 - b_2^{(\text{IV})} \gamma \qquad \forall v_1 \in \Sigma \tag{33}$$

where all the other parameters of $P_{2,2}^{(\text{IV})}$ remain the same as $P_{2,2}^{(\text{III})}$ except the following:

- $b_1^{(\text{IV})} = |\Sigma| b_1^{(\text{III})} = |\Sigma|$ (because $b_2^{(\text{III})} = 1$).

- $dep^{(\text{IV})}(\mathbf{C}_1) = \{1\}$.

- For each $v_1$: $Y_1^{(\text{IV})}(v_1) = Y_1^{(\text{III})}(v_1, \arg\min_{v_2} Y_1^{(\text{III})}(v_1, v_2))$.

Let $\mathbf{X}^{\text{opt}} = (X_1^{\text{opt}}, X_2^{\text{opt}})$ be an optimal solution of $P_{2,2}^{(\text{III})}$. Similar to Step 2, it is easy to see that $\mathbf{X}^{\text{opt}}$ is also a feasible solution of $P_{2,2}^{(\text{IV})}$: The constraint (32) can be verified similar to Step 2. Feasibility of all the other constraints of $P_{2,2}^{(\text{IV})}$ directly follows from the definition of the $P_{2,2}^{(\text{IV})}$. Also $f_{2,2}^{(\text{IV})}(\mathbf{X}^{\text{opt}}) \leq f_{2,2}^{(\text{III})}(\mathbf{X}^{\text{opt}})$. Therefore $\mathsf{Opt}(P_{2,2}^{(\text{IV})}) \leq \mathsf{Opt}(P_{2,2}^{(\text{III})})$, and hence it is sufficient to show a lower bound for $\mathsf{Opt}(P_{2,2}^{(\text{IV})})$. Note that we can directly apply constraints (33), and (32) one after another to the objective function of $P_{2,2}^{(\text{IV})}$ and obtain a lower bound for $\mathsf{Opt}(P_{2,2}^{(\text{IV})})$:

$$\mathsf{Opt}(P_{2,2}^{(\text{IV})}) \geq (1 - b_2^{(\text{IV})} \gamma)(1 - b_1^{(\text{IV})} \gamma)$$
$$\geq (1 - |\Sigma| \gamma)^2 \qquad (\text{Since } b_1^{(\text{IV})} = b_1^{(\text{IV})} = |\Sigma|)]$$
$$\geq 1 - 2|\Sigma| \gamma.$$

Because $\mathsf{Opt}(P_{2,2}^{(\text{IV})}) \leq \mathsf{Opt}(P_{2,2}^{(\text{III})}) \leq \mathsf{Opt}(P_{3,2}^{(\text{II})}) \leq \mathsf{Opt}(P_{3,2}^{(\text{I})}) \leq \mathsf{Opt}(P_{4,2}) \leq BC(P_{\mathcal{M}}[\mathbf{V}], P_{\mathcal{N}}[\mathbf{V}])$, it implies $BC(P_{\mathcal{M}}[\mathbf{V}], P_{\mathcal{N}}[\mathbf{V}]) \geq 1 - 2|\Sigma| \gamma$, and hence $H^2(P_{\mathcal{M}}[\mathbf{V}], P_{\mathcal{N}}[\mathbf{V}]) \leq 2|\Sigma| \gamma$. $\qquad \square$

# I  Reduction from General Graphs

First we define the effective parents and the c-component relation for general causal graphs.

**Definition 14** (Effective Parents $\mathbf{Pa}^+$). *Given a general causal graph $H$ and a vertex $V_i \in \mathbf{V}$, the effective parents of $V_i$, denoted by $\mathbf{Pa}^+(V_i)$, is the set of all observable vertices $V_j$ such that either $V_j$ is a parent of $V_i$ or there exists a directed path from $V_j$ to $V_i$ that contains only unobservable variables.*

**Definition 15** (c-component). *For a given general causal graph $H$, two vertices $V_i$ and $V_j$ are related by the c-component relation if (i) there exists an unobservable variable $U_k$ such that $H$ contains two paths (i) from $U_k$ to $V_i$; and (ii) from $U_k$ to $V_i$, where both the paths use only unobservable variables, or (ii) there exists another vertex $V_z \in \mathbf{V}$ such that $V_i$ and $V_z$ (and) $V_j$ and $V_z$ are related by c-component relation.*

We study Semi Markovian Bayesian Networks (SMBN)'s without any loss of generality owing to the projection of a general causal graph to a SMCG [TP02, VP90]. For a given graph $H$ they showed that there is an equivalent SMCG $G$ such that the c-component factorization and some other important properties hold. Namely,

- The set of observable nodes in $H$ and $G$ are the same.

- The topological ordering of the observable nodes in $H$ and $G$ are the same.

- The $c$-components of $H$ and $G$ are identical and the c-component factorization formula (Lemma 9 here, (20) in Lemma 2 of [TP02]) holds even for the general causal graph (See Section 5 of [TP02]). They show this based on a known previously known reduction from $H$ to $G$ [VP90]. The proof is based on the fact that for any subset $\mathbf{S} \subseteq \mathbf{V}$ of observable variables, the induced subgraphs $G[\mathbf{S}]$ and $H[\mathbf{S}]$ require the same set of conditional independence constraints.

- The parents of nodes in $G$ are the effective parents of nodes in $H$.

All the results presented in this paper depend only on the above mentioned properties. Therefore, we can reduce the given general causal graph $H$ to a SMCG $G$ using the available reduction and work with $G$, where the parents of vertices of $G$ correspond to the effective parents of the respectives vertices of $H$. Now we proceed to show the algorithm of [VP90] that preserves all the required properties mentioned above.

**Projection Algorithm of [TP02, VP90]**    For a given causal graph $H$, the projection algorithm reduces the given causal graph $H$ to a SMCG $G$ by the following procedure:

1. For each observable variable $V_i \in V$ of $H$, add an observable variable $V_i$ in $G$.

2. For each pair of observable variables $V_i, V_j \in \mathbf{V}$, if there exists a directed edge from $V_i$ to $V_j$ in H, or if there exists a *directed* path from $V_i$ to $V_j$ that contains only unobservable variables in $H$, then add a directed edge from $V_i$ to $V_j$ in $G$.

3. For each pair of observable variables $V_i, V_j \in \mathbf{V}$, if there exists an unobservable variable $U_k$ such that there exist two *directed* paths in $H$ from $U_k$ to $V_i$ and from $U_k$ to $V_j$ such that both the paths contain only the unobservable variables, then add a bi-directed edge between $V_i$ and $V_j$ in $G$.

# J    Conditional Independence

The following lemma captures a useful fact about conditional independence between variables in a SMBN.

**Lemma 17** (Independence Lemma). *Let $M$ be a SMBN with respect to a SMCG $G$ with the vertex set $\mathbf{V} = \{V_1, \ldots, V_n\}$ (where the indices respect topological ordering). For a given intervention $do(\mathbf{t})$, let $\mathbf{C} = \{V_{n_1}, V_{n_2}, \ldots, V_{n_s}\}$ be a c-component of the induced subgraph $G' = G[\mathbf{V} \setminus \mathbf{T}]$, where $s = |\mathbf{C}|$ and $n_1 < n_2 < \cdots < n_s$. Then for a given vertex $V_{n_i}$, for a given set $\mathbf{D}$ such that $\mathbf{V} \setminus (\mathbf{T} \cup \{V_{n_1}, \ldots, V_{n_i}\}) \supseteq \mathbf{D} \supseteq \mathbf{Pa}_{G'}(\{V_{n_1}, \ldots, V_{n_i}\})$, and a given set of assignments $v_{n_1}, \ldots, v_{n_i}, \mathbf{d}$,*

$$P_{\mathcal{M}}[v_{n_i} \mid v_{n_1}, \ldots, v_{n_{i-1}}, do(\mathbf{d}, \mathbf{t})] = P_{\mathcal{M}}[v_{n_i} \mid v_{n_1}, \ldots, v_{n_{i-1}}, do(\mathbf{pa}_{G'}(V_{n_1}, \ldots, V_{n_i}), \mathbf{t})]$$

*where $pa_{G'}(v_{n_1}, \ldots, v_{n_i})$ is the assignment that is consistent with $\mathbf{D}$.*

*Proof.* By Bayes' theorem

$$P_{\mathcal{M}} \left[ v_{n_{j,i}} \middle| \begin{array}{l} v_{n_{j,1}}, \ldots, v_{n_{j,i-1}}, \\ do(\mathbf{pa}_{G'}(V_{n_{j,1}}, \ldots, V_{n_{j,i}}), \mathbf{t}) \end{array} \right] \tag{34}$$

$$= \frac{P_{\mathcal{M}}[v_{n_{j,i}}, v_{n_{j,1}}, \ldots, v_{n_{j,i-1}} \mid do(\mathbf{pa}_{G'}(V_{n_{j,1}}, \ldots, V_{n_{j,i}}), \mathbf{t})]}{P_{\mathcal{M}}[v_{n_{j,1}}, \ldots, v_{n_{j,i-1}} \mid do(\mathbf{pa}_{G'}(V_{n_{j,1}}, \ldots, V_{n_{j,i}}), \mathbf{t})]}. \tag{35}$$

We apply Lemma 8 with respect to the graph $G' = G[\mathbf{V} \setminus \mathbf{T}]$ that is obtained after the intervention $do(\mathbf{t})$ for both the numerator and the denominator of (34) seperately. Therefore:

$$P_{\mathcal{M}} \left[ v_{n_{j,i}} \middle| \begin{array}{l} v_{n_{j,1}}, \ldots, v_{n_{j,i-1}}, \\ do(\mathbf{pa}_{G'}(V_{n_{j,1}}, \ldots, V_{n_{j,i}}), \mathbf{t}) \end{array} \right] = \frac{P_{\mathcal{M}}[v_{n_{j,i}}, v_{n_{j,1}}, \ldots, v_{n_{j,i-1}} \mid do(\mathbf{d}, \mathbf{t})]}{P_{\mathcal{M}}[v_{n_{j,1}}, \ldots, v_{n_{j,i-1}} \mid do(\mathbf{d}, \mathbf{t})]}$$

$$= P_{\mathcal{M}}[v_{n_{j,i}} \mid v_{n_{j,1}}, \ldots, v_{n_{j,i-1}}, do(\mathbf{d}, \mathbf{t})].$$

$\square$

## Footnotes

[7]More precisely, the goal is to discover the causal graph given the conditional independence relations satisfied by the interventional distributions.

[8]The sample complexity here is an improvement of the previously known result of [DK16].

[9] In our construction, $\mathbf{Pa}(C)$ always take $\mathbf{0}$ in the natural distribution. Henceforth, the interventions where some vertices in $\mathbf{Pa}(C)$ are not intervened are not considered here, as they are equivalent to the case when those vertices are intervened with $\mathbf{0}$.

[10]Recall that $\mathbf{pa}(\mathbf{C})$ is the assignment that gets fixed after the algorithm fixes the sequence $\mathbf{I}$.

[11]We consider the worst case, where the algorithm is provided with infinite samples.

[12]Recall that our objective is to prove: $P_{\mathcal{M}}[\mathbf{C} \setminus \mathbf{T} \mid do(\mathbf{t})] = \prod_i P_{\mathcal{M}}[\mathbf{C}_i \mid do(\mathbf{t})] = \prod_i P_{\mathcal{N}}[\mathbf{C}_i \mid do(\mathbf{t})] = P_{\mathcal{N}}[\mathbf{C} \setminus \mathbf{T} \mid do(\mathbf{t})]$, where for each $i$, $P_{\mathcal{M}}[\mathbf{C}_i \mid do(\mathbf{t})] = P_{\mathcal{N}}[\mathbf{C}_i \mid do(\mathbf{t})]$ is a uniform distribution over $\{0,1\}^{|\mathbf{C}_i|}$.

[13]We refer $\mathbf{pa}(V_j)$ with respect to the assignment $\mathbf{pa}(\mathbf{C})$.

[14]We refer to the unobservable variables $U_i$'s where both the children of $U_i$ lie in $\mathbf{C}_1$.

[15]Here, we allow some members of $\mathcal{C}$ to be empty sets.

[16] We understand that this item is very subjective.

[17] The function $Y_j(\cot) = 1$ and $b_j = 1$ take trivial values in the original program $P_{r,p}$, the $Y_j$'s and $b_j$'s will play a crucial role in the intermediate programs obtained during this process.

[18] Here the subscript $4$ denotes the number of vertices of $\tilde{G}$, and the subscript $2$ denotes the number of c-components of $\tilde{G}$.

[19] We will define a sequence of new programs while we prove the theorem, where the superscrips in roman numerals are only used to indicate the the corresponding program of the sequence.