[Reviews · NeurIPS 2018]

Reviewer 1



The paper shows some bounds on the number of interventions and samples for distinguishing causal BNs on the same graph with some unobserved nodes(?). The paper is structured and written very poorly: Introductions takes 6 pages 4 pages of citations many of which are not relevant here weird and unnecessary quotes theorems inside introduction intro includes a lot of unnecessary common knowledge to anybody actually interested in the paper e.g. two definitions of C-components results and their connections are not explained but only stated huge appendix (the submission has 31 pages in total) including most of the theoretical contribution The paper would have to be totally rewritten in order for a reviewer to sensibly form an opinion about the theoretical findings, which may themselves be good or not. AFTER AUTHOR FEEDBACK: I re-read the paper. I cannot give any higher score: the main paper is extremely poorly written, even if there would be something meaningful behind the results (which the other reviews seem to indicate). Writing a poor discussion (also, without any examples) on the results and putting a whole journal paper in the appendix is not enough. Minor: l. 190: A bidirected edge in semimarkovian graph not only put in when there is a common unobservable parent, but also when there is a common unobservable ancestor, e.g. X<-U1<-U2->U3->Y also turns into X<->Y.

Reviewer 2



The paper works on testing the interventional closeness of two causal models defined on the same causal graph. This is a non-trivial problem and authors make several key observations that may be useful for further research in the field, not only for testing, but also for learning causal graphs and causal models. This is by far the best paper in my batch and I would like to thank the authors for the well-written manuscript. Section 1.3, titled "Overview of our techniques" is especially well written and nicely summarizes the approach. There are some typos, which I believe the authors will fix in the camera ready if the paper is accepted. I pointed these out below. The authors make critical use of c-components. They leverage the fact that interventional distribution can be written in terms of the probabilities of c-components, when the remaining nodes are intervened on. Unfortunately due to space constraints, the intuition behind this factorization is not explained in depth. I believe a description on an example graph in the appendix would help make paper reach a broader audience who are unfamiliar with c-components. The fundamental contribution of the paper is to observe that the distance between any two interventional distributions can be bounded by the distances between a small number of key interventional distributions. The proof of this subadditivity result is very technical. But the key non-trivial step in the proof seems to be the lower bound given in Appendix G. This section is currently very non-intuitive. To make the results more accessible and to help with the presentation, I recommend the authors to consider a simple causal graph with latents on few variables, and write a copy of the proof of Theorem 3 using the variable names and the topological ordering of the graph. This would greatly help convey the ideas in the proof. This is partly because the notation is somewhat cumbersome in the most general form. Also, I recommend explaining the correspondence between the interventional formulation and the general formulation given in Appendix G in greater depth. For example, why are Y functions introduced in the generalized form? In the causally sufficient case, the proof is greatly simplified as given in the proof of Theorem 6 and easy to parse and verify. A set of covering interventions can be obtained by a simple randomized algorithm as given by the authors in Section E1 of the Appendix. The authors mention that it is possible to de-randomize this algorithm in Remark 1 in the Appendix, but the details are not given. It maybe useful to lay out the steps for the audience unfamiliar with de-randomization. Tian and Pearl is cited for Lemma 9, but is not cited in Eqn. (1) in the main text where the observational c-component factorization is given. Moreover, as you introduce c-component factorization here, please mention that c-components form a partitioning of the vertex set, i.e., isolated single vertices on the graph induced on bidirected edges also count as c-components. Some typos/grammar issues: Line 89: Does CGFT stand for causal goodness of fit? I could not find this abbreviation being defined. Line 100: You may want to add subscript indexes to Us and Vs Line 117: even though he doesn't know -> even though he/she does not know Line 162: causalgraphs -> causal graphs Line 163: Kacaoglu et al. -> Kocaoglu et al. (also in Line 586) Line 179: the the -> the Line 246: This seems to be hard problem. -> This seems to be a hard problem. Line 260: Let .. denotes -> Let .. denote Line 261: Size at most -> Size of at most Line 338: I does not intervene any node -> I does not intervene on any node Eqn's jump to (7) in pg 22 from (3) in pg 21 First line of (7) in Appendix: Remove the comma inside the sqrt It may be useful to mention in the text that the column-wise notation within the square-roots simply means multiplication of the terms in the square root, and is only used to be able to represent lengthy multiplications within a single line. AFTER AUTHOR FEEDBACK I would like to thank the authors for acknowledging my remarks and promising to make the changes I suggested. I believe the paper will greatly benefit from these in terms of clarity and presentation.

Reviewer 3



This paper answers a series of important questions about hypothesis testing and learning of causal models that can contain both measured and unmeasured variables, and contain only discrete variables. (The causal models are represented by semi-Markov causal models that contain a directed edge from A to B representing that A is a direct cause of B, and a bidirected edge between A and B when there is an unmeasured common cause of A and B). It provides bounds on the number of interventions, samples and time steps required by the algorithms. The distance between two models is measured by the maximum difference between interventions on the models. For an unknown model it is assumed that an experiment can perform experiments on any subset of the variables, and then measure the results of the intervention on a sample. The problems include determining when a hypothesized causal model is within epsilon of the unknown true model, when two unknown models are within epsilon of each other, and how to output a model that is within epsilon of the true model (given the correct graph). The paper also shows that the number of interventions required by their algorithms are almost optimal for the latter two problems. The fundamental strategy is to find a small "covering set" of interventions that supports a locality result that shows that when two causal models are far apart, there exists a marginal distribution of some intervention in the covering set that is far apart. I am familiar with the causal part of the arguments, but less familiar with the complexity part of the arguments. I don't understand the cases where d represents not only the maximum indegree of the model, but also the maximum degree of the model Lemma 2 states that outputting a covering set of interventions is O(K^(ld)*((3d)^l)(ld^2 log(K), where K is the number of different values the variables can take, l is the maximum size of a c-component, and d is the maximum degree of a vertex. However, there is nothing in this formula which mentions n, the number of vertices in the graph (unlike the case where d represents only the maximum indegree of the model). Similarly, Theorem 4 says that same formula is the complexity of determining when two unknown causal models are the same. But given a DAG that consists of a number of subgraphs that are disconnected from each other, and each subgraph consists of a variable and a fixed number of different parents, the number of disconnected subgraphs does not change K, l or d. Nevertheless all of the individual subgraphs would have to be experimented on, and the number of interventions needed would have to increase with n in the worst case. Also, the number of subtests performed by the algorithm is said to depend on n. There is a subtlety about the models that they are testing and finding that is not mentioned in the article. Once there are latent variables that can't be intervened on, there are multiple causal models that can generate the results of all of the possible interventions. For example, if a latent variable L causes measured X1 and X2, there are parameter values which make both X1 and X2 dependent on L, but independent of each other. In addition, intervening on X1 or X2 will produce no change in the other measured variable. So the intervention distributions can be represented by a semi-Markovian causal model in which X1 and X2 have no edge between them in the causal graph (M1) , as well as a particular parameterization of a semi-Markovian causal model in which X1 and X2 are connected by a bi-connected edge (M2). No experiment on just X1 and X2 can distinguish M1 from M2. So while the "true" causal model (i.e. the one that generated the data) might be M2, the algorithm will state that a hypothesized properly parameterized M1 is correct. For many purposes that is correct (if understood as being about correct predictions about interventions) but it is not the model that generated the data, and while M2 might suggest looking for an unmeasured common cause to manipulate, M1 does not. There were a couple of minor typos. Theorem 2 uses Sigma before it is defined. Lemma 2 refers to delta - is that the same delta as defined in Lemma 1? This paper has theoretical significance, particularly with respect to lower bounds on causal inference by any algorithm, as in Theorem 2. It is not clear what the practical significance of the algorithms or the theoretical results are for several reasons. First, most of the algorithms assume that the causal graph is completely known. Second, because the number of interventions (and sample sizes and time) grows so quickly with the size of the number of values of variables, the maximum indegree, and the size of the largest c-component. Even with l = 3, d = 4, K =4, K^(ld) which appears in the algorithm for choosing the covering set of interventions, is more than a million.